# Statistical Test for Feature Selection Pipelines by Selective Inference

**Tomohiro Shiraishi** [* 1 2] **Tatsuya Matsukawa** [* 1] **Shuichi Nishino** [1 2] **Ichiro Takeuchi** [1 2]

## Abstract

A data analysis pipeline is a structured sequence of steps that transforms raw data into meaningful insights by integrating various analysis algorithms. In this paper, we propose a novel statistical test to assess the significance of data analysis pipelines. Our approach enables the systematic development of valid statistical tests applicable to any feature selection pipeline composed of predefined components. We develop this framework based on selective inference, a statistical technique that has recently gained attention for data-driven hypotheses. As a proof of concept, we focus on feature selection pipelines for linear models, composed of three missing value imputation algorithms, three outlier detection algorithms, and three feature selection algorithms. We theoretically prove that our statistical test can control the probability of false positive feature selection at any desired level, and demonstrate its validity and effectiveness through experiments on synthetic and real data. Additionally, we present an implementation framework that facilitates testing across any configuration of these feature selection pipelines without extra implementation costs.

## 1. Introduction

In practical data-driven decision-making tasks, integrating various types of data analysis steps is crucial for addressing diverse challenges. For instance, in genetic research aimed at identifying genes linked to a specific disease, the process often begins with preprocessing tasks such as filling in missing values and detecting outliers. This is followed by screening for potentially related genes using simple descriptive statistics and then applying more complex machine learning-based feature selection algorithms. Such a systematic sequence of steps designed to analyze data and derive

useful insights is known as a *data analysis pipeline*, which plays a key role in ensuring the reproducibility and reliability of data-driven decision-making.

In this study, as an example of data analysis pipelines, we consider a class of feature selection pipelines that integrates various missing-value imputations (MVI) algorithms, outlier detection (OD) algorithms, and feature selection (FS) algorithms. Figure 1 shows examples of two such pipelines. The pipeline on the left starts with a mean value imputation algorithm, followed by $L_1$ regression based OD algorithm, proceeds with marginal screening to refine feature candidates, and concludes by using two FS algorithms—stepwise feature selection and Lasso—selecting their union as the final features. The pipeline on the right initiates with regression imputation, continues with marginal screening to narrow down feature candidates, uses Cook's distance for OD, and applies both stepwise FS and Lasso, ultimately choosing the intersection of their results as the final features.

When a data-driven approach is used for high-stakes decision-making tasks such as medical diagnosis, it is crucial to quantify the reliability of the final results by considering all steps in the pipeline. The goal of this study is to develop a statistical test for a specific class of feature selection pipelines in linear models, allowing the statistical significance of features obtained through the pipeline to be properly quantified in the form of $p$-values. The first technical challenge in achieving this is the need to appropriately account for the complex interrelations between pipeline components to determine the overall statistical significance. The second challenge is to develop a universal framework capable of performing statistical tests on arbitrary pipelines (within a given class) rather than creating individual tests for each pipeline.

To address these challenges, we introduce the concept of selective inference (SI) (Taylor & Tibshirani, 2015; Fithian et al., 2015; Lee & Taylor, 2014), a novel statistical inference approach that has gained significant attention over the past decade. The core idea of SI is to characterize the process of selecting hypotheses from the data and calculate the corresponding $p$-values using the sampling distribution, conditional on this selection process. We propose an approach based on SI that provides valid $p$-values for any feature selec-

---

*Equal contribution [1]Nagoya University, Aichi, Japan [2]RIKEN, Tokyo, Japan. Correspondence to: Ichiro Takeuchi <takeuchi.ichiro.n6@f.mail.nagoya-u.ac.jp>.

*Proceedings of the 42$^{nd}$ International Conference on Machine Learning*, Vancouver, Canada. PMLR 267, 2025. Copyright 2025 by the author(s).

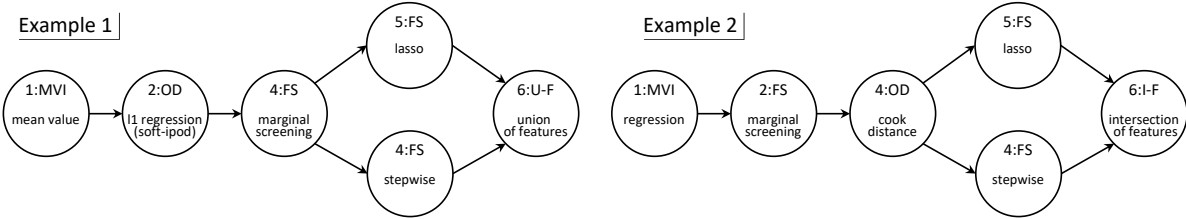

*Figure 1.* Two examples of pipelines within the class considered in this study.

tion pipeline configuration within the aforementioned class. We also introduce a modular implementation framework that supports SI for any pipeline configuration within this class without requiring additional implementation efforts. Specifically, with our framework, the statistical significance of features from any pipeline in this class can be quantified as valid $p$-values when used in a linear model, with no extra implementation required beyond specifying the pipeline.

We note that our long-term goal beyond this current study is to ensure the reproducibility of data-driven decision-making by accounting for the entire pipeline from raw data to the final results, with the current study on a class of feature selection pipelines in linear models serving as a proof of concept for that goal.

**Related Work.**  Most research on data analysis pipelines is concentrated in the field of software engineering rather than machine learning (Sugimura & Hartl, 2018; Hapke & Nelson, 2020; Drori et al., 2021), with a primary focus on the design, implementation, testing, and maintenance of pipeline systems to ensure efficiency, scalability, and robustness. Meanwhile, AutoML has emerged as a related area where researchers are automating the construction of these pipelines, and many companies have developed tools for this purpose (Microsoft, 2018; Amazon, 2019; Google, 2021). However, to the best of our knowledge, there is no existing studies that systematically discusses the reliability of data analysis pipelines. Resampling techniques, such as cross-validation (CV), are commonly used to evaluate the entire data analysis process. However, practical data analysis often includes unsupervised learning tasks like MVIs and ODs, where resampling cannot be used to accurately evaluate the reliability of the entire pipeline. Additionally, dividing the data reduces the sample size, leading to decreased accuracy in hypothesis selection and statistical power.

SI has gained attention as a statistical inference method for feature selection in linear models (Taylor & Tibshirani, 2015; Fithian et al., 2015). It has been applied to various feature selection algorithms such as marginal screening (Lee & Taylor, 2014), stepwise FS (Tibshirani et al., 2016), and Lasso (Lee et al., 2016), and extended to more complex methods (Yang et al., 2016; Suzumura et al., 2017; Hyun et al., 2018; Rügamer & Greven, 2020; Das et al., 2022;

Rügamer et al., 2022). SI is valuable not only for FS in linear models but also for inference across various data-driven hypotheses, including tasks like OD (Chen & Bien, 2020; Tsukurimichi et al., 2022), segmentation (Tanizaki et al., 2020; Duy et al., 2022; Le Duy et al., 2024), clustering (Lee et al., 2015; Gao et al., 2022), and change-point detection (Duy et al., 2020; Jewell et al., 2022). The core idea of SI is to perform statistical inference using a distribution conditioned on events of hypothesis selection, with the technical challenge being the characterization of various event selections for different tasks. While studies on SI for various tasks are being conducted, existing research is limited to single tasks, and how to perform inference when integrating multiple tasks into a pipeline remains an open question. Furthermore, existing implementations of SI are developed individually for each task, and there is no unified framework for implementing SI.

**Contributions.**  Our contributions in this study are threefold. First, we develop a statistical test for feature selection pipelines composed of various configurations of missing value imputation (MVI), outlier detection (OD), and feature selection (FS) components, based on the SI framework. Second, this study represents the first application of SI to inference on a combination of multiple analysis components in a unified, systematic manner. Finally, we provide a practical computational framework implemented as the Python package[1], which facilitates the construction of statistical tests across any pipeline configuration without additional implementation costs.  For reproducibility, our experimental code is available at `https://github.com/shirara1016/statistical_test_for_feature_selection_pipelines`.

## 2. Preliminaries

Given a set of algorithm components, a pipeline is defined by selecting some components from the set and connecting the selected components in an appropriate way. A pipeline can be represented as a directed acyclic graph (DAG) with components as nodes, and the connections as edges. In this study, as an example class of pipelines, we consider a set of

---

[1]`https://pypi.org/project/si4pipeline/`

algorithms consisting of three MVI algorithms, three OD algorithms, three FS algorithms, as well as *Intersection* and *Union* operations (specific three algorithms each for MVI, OD, and FS are described later in this section). Figure 1 shows two examples of pipelines within this class. Note that each FS algorithm corresponds to a single node in the DAG, and the FS algorithms are not described as DAGs.

**Problem Setting.** In this study, we consider the problem of feature selection for linear models from a dataset containing missing values and/or outliers using the aforementioned class of feature selection pipelines. Let us consider a linear regression problem with $n$ instances and $d$ features. We denote the observed dataset as $(X, \boldsymbol{y})$, where $X \in \mathbb{R}^{n \times d}$ is the fixed design matrix, while $\boldsymbol{y} \in \mathbb{R}^{n'}$ is the response vector which contains outlying values but excludes missing values (i.e., $n' \leq n$). We assume that $\boldsymbol{y}$ is a random realization of the following random response vector

$$\boldsymbol{Y} = \boldsymbol{\mu}(X) + \boldsymbol{\varepsilon}, \ \boldsymbol{\varepsilon} \sim \mathcal{N}(\boldsymbol{0}, \sigma^2 I_{n'}), \tag{1}$$

where $\boldsymbol{\mu}(X) \in \mathbb{R}^{n'}$ is the unknown true value function, while $\boldsymbol{\varepsilon} \in \mathbb{R}^{n'}$ is independently normally distributed with variance $\sigma^2$ which is known or estimable from an independent dataset[2]. Although we do not pose any functional form on the true value function $\boldsymbol{\mu}(X)$ for theoretical justification, we consider a case where the true values $\boldsymbol{\mu}(X)$ are reasonably approximated by a linear model as long as they are non-outliers. This is a common setting in the field of SI, referred to as the *saturated model* setting. Furthermore, we denote the response vector with imputed missing values as $\boldsymbol{y}^{(+)} \in \mathbb{R}^n$. Using the above notations, a feature selection pipeline comprising of MVI, OD, and FS algorithm components is represented as a mapping:

$$\mathcal{P} : \mathbb{R}^{n \times d} \times \mathbb{R}^{n'} \ni (X, \boldsymbol{y}) \mapsto (\boldsymbol{y}^{(+)}, \mathcal{O}, \mathcal{M}) \in \mathbb{R}^n \times 2^{[n]} \times 2^{[d]}, \tag{2}$$

where $\boldsymbol{y}^{(+)} \in \mathbb{R}^n$ is the response vector with missing values imputed, $\mathcal{O} \subset [n]$ is the set of detected outliers, and $\mathcal{M} \subset [d]$ is the set of selected features.

**Statistical Test for Pipelines.** Given the output of a pipeline in (2), the statistical significance of the finally selected features can be quantified based on the coefficients of the linear model fitted only with the selected features from a dataset with missing values imputed and outliers removed. To formalize this, we denote the design matrix after removing outliers and composed only of the selected features as $X_{-\mathcal{O}, \mathcal{M}} \in \mathbb{R}^{n - |\mathcal{O}| \times |\mathcal{M}|}$, and denote the response vector with outliers removed and missing values imputed as $\boldsymbol{y}_{-\mathcal{O}}^{(+)}$. Using these notations, the least squares solution of

—————
[2]We discuss the robustness of the proposed method when the variance is unknown and the noise deviates from the Gaussian distribution in Appendix E.

the linear model after imputation of missing values, removal of outliers, and feature selection is expressed as

$$\hat{\boldsymbol{\beta}} = \left( X_{-\mathcal{O}, \mathcal{M}}^{\top} X_{-\mathcal{O}, \mathcal{M}} \right)^{-} X_{-\mathcal{O}, \mathcal{M}}^{\top} \boldsymbol{y}_{-\mathcal{O}}^{(+)}.$$

Similarly, we consider the population least-square solution for the unobservable true value vector $\boldsymbol{\mu}(X)$ in (1), which is defined as

$$\boldsymbol{\beta}^* = \left( X_{-\mathcal{O}, \mathcal{M}}^{\top} X_{-\mathcal{O}, \mathcal{M}} \right)^{-} X_{-\mathcal{O}, \mathcal{M}}^{\top} \boldsymbol{\mu}_{-\mathcal{O}}^{(+)}(X_{-\mathcal{O}, \mathcal{M}}),$$

where $\boldsymbol{\mu}_{-\mathcal{O}}^{(+)}(X_{-\mathcal{O}, \mathcal{M}}) \in \mathbb{R}^{n - |\mathcal{O}|}$ is an $n - |\mathcal{O}|$-dimensional vector obtained by providing $X_{-\mathcal{O}, \mathcal{M}}$ to the unknown true function $\boldsymbol{\mu}$ with the missing values imputed with the same MVI algorithm. To quantify the statistical significance of the selected features, we consider the following null hypothesis $H_0$ and the alternative hypothesis $H_1$:

$$H_0 : \beta_j^* = 0 \text{ v.s. } H_1 : \beta_j^* \neq 0, \ j \in \mathcal{M}, \tag{3}$$

where, with a slight abuse of notation, $\beta_j^*$ and $\hat{\beta}_j$ respectively indicates the element of $\boldsymbol{\beta}^*$ and $\hat{\boldsymbol{\beta}}$ corresponding to the selected feature $j \in \mathcal{M}$.

**Missing-Value Imputation (MVI) Algorithm Components.** In this paper, as three examples of MVI algorithms, we consider *mean value imputation*, *nearest-neighbor imputation*, and *regression imputation* algorithms (see Appendix A.1). A MVI algorithm component is represented as

$$f_{\mathrm{MVI}} : \{X, \boldsymbol{y}, \mathcal{O}, \mathcal{M}\} \mapsto \{X, \boldsymbol{y}^+, \mathcal{O}, \mathcal{M}\},$$

where, among the four variables, only $\boldsymbol{y}$ is updated to $\boldsymbol{y}^{(+)}$, but note that this notation is used to uniformly handle all components in the pipeline. It is important to note that these three MVI algorithms are *linear* algorithm in the sense that, using a matrix $D_X \in \mathbb{R}^{n \times n'}$ that depends on $X$, the imputed values are written as $\boldsymbol{y}^{(+)} = D_X \boldsymbol{y}$.

**Outlier Detection (OD) Algorithm Components.** In this paper, as three examples of OD algorithms, we consider *Cook's distance-based OD*, *DFFITS OD*, and $L_1$ *regression based OD* algorithms (see Appendix A.2). A OD algorithm component is represented as

$$f_{\mathrm{OD}} : \{X, \boldsymbol{y}^{(+)}, \mathcal{O}, \mathcal{M}\} \mapsto \{X, \boldsymbol{y}^{(+)}, \mathcal{O}', \mathcal{M}\},$$

where, $\mathcal{O}'$ is the updated set of outliers. Note that, if outlier removal and feature selection have not yet been performed, the sets $\mathcal{O}$ and $\mathcal{M}$ are initialized as $\mathcal{O} = \emptyset$ and $\mathcal{M} = [d]$.

**Feature Selection (FS) Algorithm Components.** In this paper, as three examples of FS algorithms, we consider *marginal screening*, *stepwise feature selection*, and *Lasso* algorithms (see Appendix A.3). A FS algorithm component is represented as

$$f_{\mathrm{FS}} : \{X, \boldsymbol{y}^{(+)}, \mathcal{O}, \mathcal{M}\} \mapsto \{X, \boldsymbol{y}^{(+)}, \mathcal{O}, \mathcal{M}'\},$$

where, $\mathcal{M}'$ is the updated set of features.

**Union and Intersection Components.** When using multiple OD/FS algorithms, it is necessary to include components in the pipeline that perform the union/intersection of the detected outliers or selected features. Such union/intersection components for OD/FS are respectively written as

$$f_\Sigma^{\mathcal{O}} : \{X, \boldsymbol{y}^{(+)}, \{\mathcal{O}_e\}_{e \in [E]}, \mathcal{M}\} \mapsto \{X, \boldsymbol{y}^{(+)}, \Sigma_{e \in [E]} \mathcal{O}_e, \mathcal{M}\},$$

$$f_\Sigma^{\mathcal{M}} : \{X, \boldsymbol{y}^{(+)}, \mathcal{O}, \{\mathcal{M}_e\}_{e \in [E]}\} \mapsto \{X, \boldsymbol{y}^{(+)}, \mathcal{O}, \Sigma_{e \in [E]} \mathcal{M}_e\},$$

where $E$ is the number of OD/FS algorithms and an operator $\Sigma$ indicates either union or intersection of multiple sets.

**Automatic Pipeline Construction.** In this study, we consider two cases for pipeline configuration: an option specified by the user and an option determined based on the data. In the first option, the user can select some of the aforementioned data analysis components and specify their own configuration. On the other hand, the second option allows for the selection of the optimal configuration from among multiple pre-defined pipeline configurations based on CV. An important point in the second option is that our statistical test is designed by properly considering the fact that the optimal pipeline configuration has been selected based on the data[3]. For more details on the second option, see §6 and Appendix F.

**Selective Inference.** For the statistical test in (3), it is reasonable to use $\hat{\beta}_j, j \in \mathcal{M}$ as the test statistic. An important point when addressing this statistical test within the SI approach is that the test statistic is represented as a linear function of the observed response vector as $\hat{\beta}_j = \boldsymbol{\eta}_j^\top \boldsymbol{y}, j \in \mathcal{M}$, where $\boldsymbol{\eta}_j \in \mathbb{R}^{n'}, j \in \mathcal{M}$ is a vector that depends on $\boldsymbol{y}$ only through the detected outlier set $\mathcal{O}$ and the selected feature set $\mathcal{M}$ [4]. In SI, this property is utilized to perform statistical inference based on the sampling distribution of the test statistic conditional on $\mathcal{O}$ and $\mathcal{M}$. More specifically, since $\boldsymbol{y}$ follows a normal distribution, it can be derived that the sampling distribution of the test statistic $\hat{\beta}_j = \boldsymbol{\eta}_j^\top \boldsymbol{y}, j \in \mathcal{M}$ conditional on $\mathcal{O}, \mathcal{M}$, and the sufficient statistic of the nuisance parameters follows a truncated normal distribution. By computing $p$-values based on this conditional sampling distribution represented as a truncated normal distribution, it is ensured that the type I error can be controlled even in finite samples. For more details on SI, please refer to the following sections or literatures such as Taylor & Tibshirani (2015); Fithian et al. (2015); Lee & Taylor (2014).

---

[3]As stated in §1, CV cannot be used for an accurate evaluation of a pipeline when it includes unsupervised learning components such as MVI or OD. However, it is possible to compute a valid $p$-value for a pipeline selected by CV if we properly consider the CV-based pipeline selection as part of the selection event for SI.

[4]Note that the MVI algorithms considered in this paper depend only on $X$, not on $\boldsymbol{y}$.

## 3. Selective Inference for Feature Selection Pipelines

To perform statistical test for pipelines, it is necessary to consider how the data influenced the final result through the calculations of each algorithm component of the pipeline and in operations where they are combined with a specified configuration. We address this challenge using the SI framework. In the SI, statistical inference is performed based on the sampling distribution conditional on the process by which the data selects the final result, thereby incorporating the influence of how data is processed in the pipeline.

**Selective Inference.** In SI, $p$-values are computed based on the null distribution conditional on an event that a certain hypothesis is selected. The goal of SI is to compute a $p$-value such that

$$\mathbb{P}_{\mathrm{H}_0} (p \le \alpha \mid \mathcal{M}_{\boldsymbol{Y}} = \mathcal{M}, \mathcal{O}_{\boldsymbol{Y}} = \mathcal{O}) = \alpha, \ \forall \alpha \in (0, 1), \tag{4}$$

where $\mathcal{M}_{\boldsymbol{Y}}$ and $\mathcal{O}_{\boldsymbol{Y}}$ respectively indicate the random set of selected features and detected outliers given the random response vector $\boldsymbol{Y}$, thereby making the $p$-value is a random variable. Here, the condition part $\mathcal{M}_{\boldsymbol{Y}} = \mathcal{M}$ and $\mathcal{O}_{\boldsymbol{Y}} = \mathcal{O}$ in (4) indicates that we only consider response vectors $\boldsymbol{Y}$ yielding a certain feature set $\mathcal{M}$ and a certain outlier set $\mathcal{O}$. If the conditional type I error rate can be controlled as in (4) for any possible hypotheses $(\mathcal{M}, \mathcal{O}) \in 2^{[d]} \times 2^{[n]}$, then, by the law of total probability, the marginal type I error rate can also be controlled for any $\alpha \in (0, 1)$ because

$$\mathbb{P}_{\mathrm{H}_0}(p \le \alpha)$$
$$= \sum_{\mathcal{M} \in 2^{[d]}} \sum_{\mathcal{O} \in 2^{[n]}} \begin{matrix} \mathbb{P}_{\mathrm{H}_0}(\mathcal{M}, \mathcal{O}) \\ \mathbb{P}_{\mathrm{H}_0} (p \le \alpha \mid \mathcal{M}_{\boldsymbol{Y}} = \mathcal{M}, \mathcal{O}_{\boldsymbol{Y}} = \mathcal{O}) \end{matrix}$$
$$= \alpha.$$

Therefore, in order to perform valid statistical test, we can employ $p$-values conditional on the hypothesis selection event. To compute a $p$-value that satisfies (4), we need to derive the sampling distribution of the test-statistic

$$T(\boldsymbol{Y}) \mid \{\mathcal{M}_{\boldsymbol{Y}} = \mathcal{M}_{\boldsymbol{y}}, \mathcal{O}_{\boldsymbol{Y}} = \mathcal{O}_{\boldsymbol{y}}\}. \tag{5}$$

**Selective $p$-value.** To conduct statistical hypothesis testing based on the conditional sampling distribution in (5), we introduce an additional condition on the sufficient statistic of the nuisance parameter $\mathcal{Q}_{\boldsymbol{Y}}$, defined as

$$\mathcal{Q}_{\boldsymbol{Y}} = \left( I_{n'} - \frac{\boldsymbol{\eta} \boldsymbol{\eta}^\top}{\|\boldsymbol{\eta}\|^2} \right) \boldsymbol{Y}. \tag{6}$$

This additional conditioning on $\mathcal{Q}_{\boldsymbol{Y}}$ is a standard practice in the SI literature required for computational tractability[5].

---

[5]The nuisance component $\mathcal{Q}_{\boldsymbol{Y}}$ corresponds to the component $\boldsymbol{z}$ in the seminal paper (Lee et al., 2016) (see Sec. 5, Eq. (5.2), and

Based on the additional conditioning on $\mathcal{Q}_{\boldsymbol{Y}}$, the following theorem tells that the conditional $p$-value that satisfies (4) can be derived by using a truncated normal distribution.

**Theorem 3.1.** *Consider a constant design matrix $X$, a random response vector $\boldsymbol{Y} \sim \mathcal{N}(\boldsymbol{\mu}, \sigma^2 I_{n'})$ and an observed response vector $\boldsymbol{y}$. Let $(\mathcal{M}_{\boldsymbol{Y}}, \mathcal{O}_{\boldsymbol{Y}})$ and $(\mathcal{M}_{\boldsymbol{y}}, \mathcal{O}_{\boldsymbol{y}})$ be the pairs of selected features and detected outliers, obtained by applying a pipeline process $\mathcal{P}$ in the form of (2) to $(X, \boldsymbol{Y})$ and $(X, \boldsymbol{y})$, respectively. Let $\boldsymbol{\eta} \in \mathbb{R}^{n'}$ be a vector depending on $(\mathcal{M}_{\boldsymbol{y}}, \mathcal{O}_{\boldsymbol{y}})$, and consider a test-statistic in the form of $T(\boldsymbol{Y}) = \boldsymbol{\eta}^{\top}\boldsymbol{Y}$. Furthermore, define the nuisance parameter $\mathcal{Q}_{\boldsymbol{Y}}$ as in (6).*

*Then, the conditional distribution*

$$T(\boldsymbol{Y}) \mid \{\mathcal{M}_{\boldsymbol{Y}} = \mathcal{M}_{\boldsymbol{y}}, \mathcal{O}_{\boldsymbol{Y}} = \mathcal{O}_{\boldsymbol{y}}, \mathcal{Q}_{\boldsymbol{Y}} = \mathcal{Q}_{\boldsymbol{y}}\}$$

*is a truncated normal distribution $\mathrm{TN}(\boldsymbol{\eta}^{\top}\boldsymbol{\mu}, \sigma^2\|\boldsymbol{\eta}\|^2, \mathcal{Z})$ with mean $\boldsymbol{\eta}^{\top}\boldsymbol{\mu}$, variance $\sigma^2\|\boldsymbol{\eta}\|^2$, and truncation intervals $\mathcal{Z}$, where $\mathcal{Z}$ is defined as*

$$\mathcal{Z} = \{z \in \mathbb{R} \mid \mathcal{M}_{\boldsymbol{a}+\boldsymbol{b}z} = \mathcal{M}_{\boldsymbol{y}}, \mathcal{O}_{\boldsymbol{a}+\boldsymbol{b}z} = \mathcal{O}_{\boldsymbol{y}}\}, \quad (7)$$
$$\boldsymbol{a} = \mathcal{Q}_{\boldsymbol{y}}, \ \boldsymbol{b} = \boldsymbol{\eta}/\|\boldsymbol{\eta}\|^2.$$

The proof of Theorem 3.1 is deferred to Appendix B.1. By using the sampling distribution of the test statistic $T(\boldsymbol{Y})$ conditional on $\mathcal{M}_{\boldsymbol{Y}} = \mathcal{M}_{\boldsymbol{y}}, \mathcal{O}_{\boldsymbol{Y}} = \mathcal{O}_{\boldsymbol{y}}$, and $\mathcal{Q}_{\boldsymbol{Y}} = \mathcal{Q}_{\boldsymbol{y}}$ in Theorem 3.1, we can define the selective $p$-value as

$$p_{\text{selective}} = \mathbb{P}_{\mathrm{H}_0}\left(|T(\boldsymbol{Y})| \geq |T(\boldsymbol{y})| \left| \begin{array}{c} \mathcal{M}_{\boldsymbol{Y}} = \mathcal{M}_{\boldsymbol{y}}, \\ \mathcal{O}_{\boldsymbol{Y}} = \mathcal{O}_{\boldsymbol{y}}, \\ \mathcal{Q}_{\boldsymbol{Y}} = \mathcal{Q}_{\boldsymbol{y}} \end{array} \right.\right). \quad (8)$$

**Theorem 3.2.** *The selective $p$-value defined in (8) satisfies the property in (4), i.e.,*

$$\mathbb{P}_{\mathrm{H}_0}\left(p_{\text{selective}} \leq \alpha \left| \begin{array}{c} \mathcal{M}_{\boldsymbol{Y}} = \mathcal{M}_{\boldsymbol{y}}, \\ \mathcal{O}_{\boldsymbol{Y}} = \mathcal{O}_{\boldsymbol{y}} \end{array} \right.\right) = \alpha, \ \forall \alpha \in (0, 1).$$

*Then, the selective $p$-value also satisfies the following property of a valid $p$-value:*

$$\mathbb{P}_{\mathrm{H}_0}(p_{\text{selective}} \leq \alpha) = \alpha, \ \forall \alpha \in (0, 1).$$

The proof of Theorem 3.2 is deferred to Appendix B.2. This theorem guarantees that the selective $p$-value is uniformly distributed under the null hypothesis $\mathrm{H}_0$, and thus can be used to conduct the valid statistical inference in (3). Once the truncation intervals $\mathcal{Z}$ is identified, the selective $p$-value in (8) can be easily computed using Theorem 3.1. Thus, the remaining task is reduced to identifying the truncation intervals $\mathcal{Z}$.

---

Theorem 5.2) and is used in almost all the SI-related works that we cited.

# 4. Computations: Line Search Interpretation

From the discussion in §3, it is suffice to identify the one-dimensional subset $\mathcal{Z}$ in (7) to conduct the inference. In this section, we propose a novel line search method to efficiently identify the $\mathcal{Z}$.

## 4.1. Overview of the Line Search

The difficulty in identifying the $\mathcal{Z}$ arises from the fact that the multiple FS/OD algorithms are applied in an arbitrary complex order. To surmount this difficulty, we propose an efficient search method that leverages parametric-programming and the fact that our pipeline can be conceptualized as a directed acyclic graph (DAG) whose nodes represent the operations. In a standard analysis pipeline, $\mathcal{M}$ and $\mathcal{O}$ are computed and updated along the DAG. However, in our framework, intervals for which $\mathcal{M}$ and $\mathcal{O}$ are constant can also be computed and updated, allowing the computation of the truncation intervals $\mathcal{Z}$.

In the following, we first discuss how, given a certain computational procedure (combining *update rules* as discussed in later), the $\mathcal{Z}$ can be identified by parametric-programming. Then, we summarize the overall procedure to compute the selective $p$-value from the $\mathcal{Z}$. Finally, we describe the update rules for each node based on the existing methods of SI for each FS and OD algorithm. Note that DAGs can topologically sortable, so that update rules can be applied in sequence. The overview of the proposed line search method is illustrated in Figure 2.

## 4.2. Parametric-Programming

To identify the truncation intervals $\mathcal{Z}$, we assume that we have a procedure to compute the interval $[L_z, U_z]$ for any $z \in \mathbb{R}$, which satisfies

$$\forall r \in [L_z, U_z], \ \mathcal{M}_{\boldsymbol{a}+\boldsymbol{b}r} = \mathcal{M}_{\boldsymbol{a}+\boldsymbol{b}z}, \mathcal{O}_{\boldsymbol{a}+\boldsymbol{b}r} = \mathcal{O}_{\boldsymbol{a}+\boldsymbol{b}z}.$$

Then, the truncation intervals $\mathcal{Z}$ can be obtained by the union of the intervals $[L_z, U_z]$ as

$$\mathcal{Z} = \bigcup_{z \in \mathbb{R} \mid \mathcal{M}_{\boldsymbol{a}+\boldsymbol{b}z} = \mathcal{M}_{\boldsymbol{y}}, \mathcal{O}_{\boldsymbol{a}+\boldsymbol{b}z} = \mathcal{O}_{\boldsymbol{y}}} [L_z, U_z]. \quad (9)$$

The procedure in (9) is commonly referred to as parametric-programming. We discuss the details of the procedure to compute the interval $[L_z, U_z]$ by defining the update rules for each node in the next subsection.

## 4.3. Update Rules

In this subsection, we discuss the computation procedure to obtain the interval $[L_z, U_z]$ for any $z \in \mathbb{R}$ just mentioned in §4.2. To compute the interval $[L_z, U_z]$, we consider extending the input of each node in a DAG and denote it

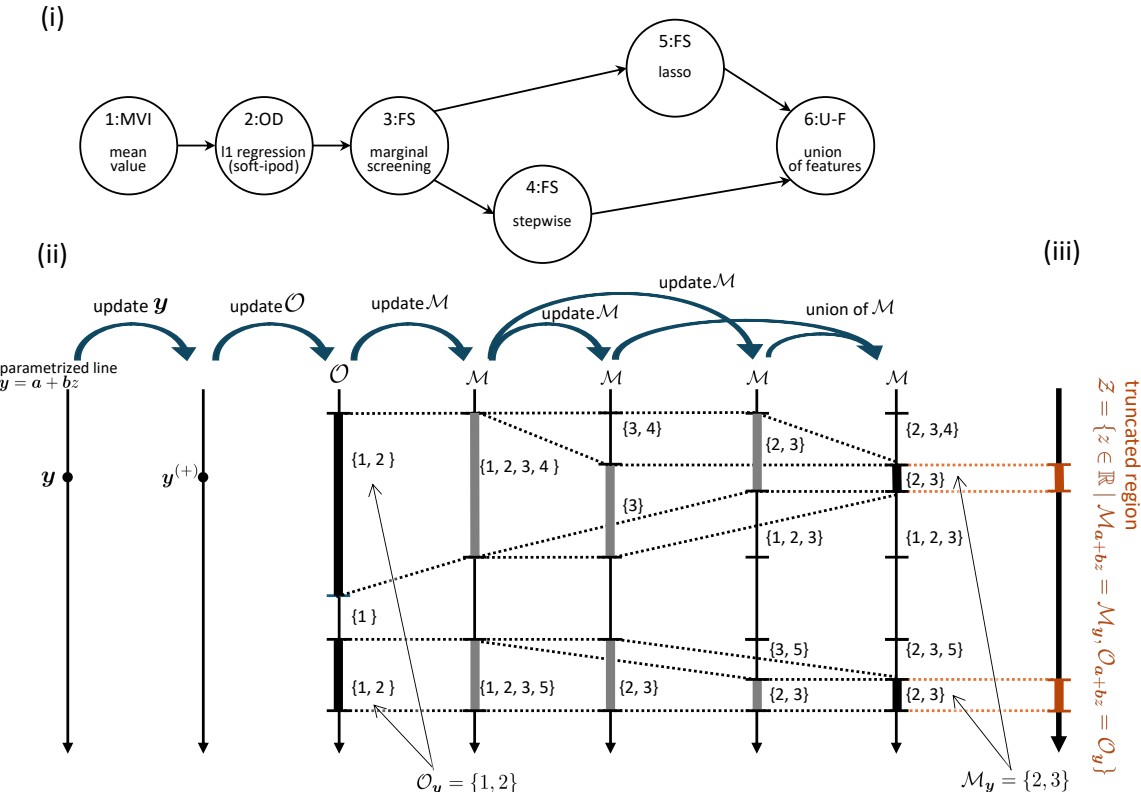

Figure 2. Schematic illustration of the proposed line search method to identify the truncation intervals $\mathcal{Z}$. The upper part shows the DAG representation of the pipeline and its topological sorting (i). The lower left part shows the operations performed by update rules in sequence (ii). The lower right part shows the identification of the truncation intervals $\mathcal{Z}$ by taking the union of some intervals based on parametric-programming (iii).

as a pair of $(X, \boldsymbol{a}, \boldsymbol{b}, z, \mathcal{M}, \mathcal{O}, l, u)$, where $X$ is the design matrix, $\boldsymbol{a}$, $\boldsymbol{b}$ and $z$ are the currently linear expression of the response vector $\boldsymbol{a} + \boldsymbol{b}z$, $\mathcal{M}$ and $\mathcal{O}$ are the currently selected features and detected outliers, and $l$ and $u$ are the currently interval. The input of the first node is initialized to $(X, \boldsymbol{a}, \boldsymbol{b}, z, [d], \emptyset, -\infty, \infty)$, where $d$ is the number of features. We details the update rules for this pair at each node of a DAG in Appendix C.

The overall procedure for computing the interval $[L_z, U_z]$ by applying the update rules in the order of the topological sorting of the DAG is summarized in Algorithm 1, where the operation pa receives the index of the target node and returns the indexes of its parent nodes, and $\mathrm{pa}(1)$ is set to 0. Algorithm 1 satisfies the specifications described in §4.2, i.e., the following theorem holds.

**Theorem 4.1.** *Consider a pipeline $\mathcal{P}$, a design matrix $X$, and vectors $\boldsymbol{a}$ and $\boldsymbol{b}$ representing the linear expression of the response vector as fixed. For any $z \in \mathbb{R}$, let $[L_z, U_z]$, $\mathcal{M}_{\boldsymbol{a}+\boldsymbol{b}z}$ and $\mathcal{O}_{\boldsymbol{a}+\boldsymbol{b}z}$ be the output of Algorithm 1 with $\mathcal{P}$, $X$, $\boldsymbol{a}$, $\boldsymbol{b}$ and $z$ as input.*

*Then, for any $r \in [L_z, U_z]$, the output of Algorithm 1 does*

*not change by changing the input $z$ to $r$:*

$$\mathtt{UpdateInterval}(\mathcal{P}, X, \boldsymbol{a}, \boldsymbol{b}, r)$$
$$= ([L_z, U_z], \mathcal{M}_{\boldsymbol{a}+\boldsymbol{b}z}, \mathcal{O}_{\boldsymbol{a}+\boldsymbol{b}z}).$$

The proof of Theorem 4.1 is deferred to Appendix B.3.

---

**Algorithm 1** Apply Update Rules in Order of Topological Sorting of DAG (Update Interval)

---

**Require:** $\mathcal{P}$, $X$, $\boldsymbol{a}$, $\boldsymbol{b}$ and $z$
1: Converts the pipeline $\mathcal{P}$ to a topologically sorted graph $(V, E)$
2: Initialize the input of the first node $B_0$ as $(X, \boldsymbol{a}, \boldsymbol{b}, z, [p], \emptyset, -\infty, \infty)$ (see §4.3)
3: **for** each index of node $i \in \{1, \dots, |V|\}$ **do**
4:     Apply the update rule of the node $v_i$ to its input $B_{\mathrm{pa}(i)}$ to obtain the output $B_i$ (see §4.3)
5: **end for**
6: Let the last four components of $B_{|V|}$ be $\mathcal{M}_{\boldsymbol{a}+\boldsymbol{b}z}$, $\mathcal{O}_{\boldsymbol{a}+\boldsymbol{b}z}$, $L_z$ and $U_z$, respectively
**Ensure:** $[L_z, U_z]$, $\mathcal{M}_{\boldsymbol{a}+\boldsymbol{b}z}$ and $\mathcal{O}_{\boldsymbol{a}+\boldsymbol{b}z}$

---

# 5. Implementations: Auto-Conditioning

All of the update rules defined in §4.3 are node-specific operations and do not depend on the type of node corresponding to the input/output. Then, we can modularize the update rules and apply them sequentially as in Algorithm 1, which implementation we call *auto-conditioning*. The auto-conditioning allows one to simply define an arbitrary pipeline and perform hypothesis testing on it without additional implementation costs. In this section, we show some examples of defining pipelines and performing hypothesis testing using the auto-conditioning. The implementation we developed can be interactively executed using the provided Jupyter Notebook (ipynb) file, which is available in the our package repository.

As an example, Listing 1 shows a code example that defines two pipeline shown in Figure 2 and performs hypothesis testing, based on our package. A similarly simple UI allows for easy implementation of other pipeline structures as well as automatic pipeline construction based on the cross-validation. For more examples, please refer to the Appendix G and the our package repository.

*Listing 1.* Code example that defines the pipeline shown in Figure 2. We can create an instance of manager class which handles the desired pipeline simply by specifying each operation in turn. To perform hypothesis testing, we can call the `inference` method of the manager instance with the input dataset $(X, \boldsymbol{y})$ and the deviation of the noise $\sigma$.

```python
import numpy as np
from si4pipeline import *

def option1() -> PipelineManager:
    X, y = initialize_dataset()
    y = mean_value_imputation(X, y)

    O = soft_ipod(X, y, 0.02)
    X, y = remove_outliers(X, y, O)

    M = marginal_screening(X, y, 5)
    X = extract_features(X, M)

    M1 = stepwise_feature_selection(X, y, 3)
    M2 = lasso(X, y, 0.08)
    M = union(M1, M2)
    return construct_pipelines(output=M)

def option2() -> PipelineManager:
    X, y = initialize_dataset()
    y = definite_regression_imputation(X, y)

    M = marginal_screening(X, y, 5)
    X = extract_features(X, M)

    O = cook_distance(X, y, 3.0)
    X, y = remove_outliers(X, y, O)

    M1 = stepwise_feature_selection(X, y, 3)
    M2 = lasso(X, y, 0.08)
    M = intersection(M1, M2)
    return construct_pipelines(output=M)

pl = option1()
X = np.random.normal(size=(100, 10))
y = np.random.normal(size=100)
M, p_list = pl.inference(X, y, sigma=1.0)
```

# 6. Numerical Experiments

**Methods for Comparison.** In our experiments, we consider the three types of pipelines: `op1`, `op2`, and `cv`. The `op1` and `op2` are defined in Figure 2. The `cv` is a pipeline selected based on cross-validation from 16 different parameters settings each in the `op1` and `op2` pipelines (i.e., from 32 pipelines in total). For each three types of pipelines, we compare the proposed method (`proposed`) in terms of type I error rate and power with the following three methods:

- `w/o-pp`: An ablation study that excludes the parametric programming technique described in §4.2. This is implemented by replacing the $\mathcal{Z}$ in (9) with a interval $[L_z, U_z]$ that contains the observed test statistic $T(\boldsymbol{y})$.

- `naive`: This method uses a classical $z$-test without conditioning, i.e., we compute the naive $p$-value as $p_{\text{naive}} = \mathbb{P}_{\text{H}_0}(|T(\boldsymbol{Y})| \geq |T(\boldsymbol{y})|)$.

- `bonferroni`: This is a method to control the type I error rate by using the Bonferroni correction, a simple yet widely used method for multiple testing correction. The number of all possible pair of selected features and detected outliers is $2^d \cdot 2^n$, then we compute the Bonferroni corrected $p$-value as $p_{\text{bonferroni}} = \min(1, 2^d \cdot 2^n \cdot p_{\text{naive}})$.

**Experimental Setup.** In all experiments, we set the significance level $\alpha = 0.05$. For the experiments to see the type I error rate, we change the number of samples $n \in \{100, 200, 300, 400\}$ and set the number of features $d$ to 20. See Appendix D.1 for results when the number of features $d$ is changed, and for the high-dimensional regression setting (i.e., where $d \gg n$). For each configuration, we generated 10,000 null datasets $(X, \boldsymbol{y})$, where $X_{ij} \sim \mathcal{N}(0, 1)$ for all $(i, j) \in [n] \times [d]$ and $\boldsymbol{y} \sim \mathcal{N}(0, I_n)$. Missing values were introduced by randomly setting each $y_i$ to NaN with a probability of 0.03. To investigate the power, we set $n = 200$ and $d = 20$ and generated dataset $(X, \boldsymbol{y})$, where $X_{ij} \sim \mathcal{N}(0, 1)$ for all $(i, j) \in [n] \times [d]$ and $\boldsymbol{y} = X\boldsymbol{\beta} + \boldsymbol{\epsilon}$. The error term $\boldsymbol{\epsilon}$ followed a normal distribution $\mathcal{N}(0, I_n)$, and the coefficient vector $\boldsymbol{\beta} \in \mathbb{R}^d$ was constructed such that its first three elements were set to $\Delta$ and the remaining elements were set to 0. Missing values were introduced by randomly setting each $y_i$ to NaN with a probability of 0.03. We change the true coefficients $\Delta \in \{0.2, 0.4, 0.6, 0.8\}$. For power evaluation, hypothesis testing was conducted only when the pipeline selected at least one truly relevant feature (i.e., one of the first three features), resulting in a total of 10,000 tests. In addition, see Appendix D.2 for results when the missing value probability increased, Appendix D.3 for the computational time of the proposed method for larger datasets and more complex pipelines, and Appendix D.4 for the computer resources used in the experiments.

*Table 1.* Power on eight real-world datasets when changing the sample size $n$. Each cell indicates the power of the proposed method (`proposed`) and the ablation study (`w/o-pp`), separated by a slash, with the higher value in bold. The proposed method demonstrates significantly higher power than the ablation study method across all datasets and sample sizes. Furthermore, the power of the proposed method increases with increasing sample size $n$.

| $n$ | Data1 | Data2 | Data3 | Data4 | Data5 | Data6 | Data7 | Data8 |
|---|---|---|---|---|---|---|---|---|
| 100 | **.57**/.07 | **.48**/.06 | **.57**/.07 | **.51**/.07 | **.68**/.10 | **.55**/.04 | **.30**/.05 | **.25**/.06 |
| 150 | **.79**/.09 | **.71**/.08 | **.66**/.12 | **.57**/.10 | **.74**/.12 | **.72**/.06 | **.37**/.06 | **.37**/.06 |
| 200 | **.91**/.11 | **.80**/.08 | **.78**/.15 | **.66**/.12 | **.76**/.13 | **.82**/.08 | **.49**/.07 | **.40**/.06 |

**Results.** The results of type I error rate are shown in left side of Figure 3. The `proposed`, `w/o-pp`, and `bonferroni` successfully controlled the type I error rate under the significance level across all settings and pipeline types, whereas the `naive` could not. Because the `naive` failed to control the type I error rate, we no longer consider its power. The results of power are shown in right side of Figure 3. Among the methods that controlled the type I error rate, the `proposed` has the highest power, followed by the `w/o-pp`, across all settings and pipeline types. The reduced power of the `w/o-pp` compared to the `proposed` can be attributed to its inherent conditioning on more information than those defined in (5). This problem is known as *over-conditioning* in the context of SI. The notably low power of the `bonferroni` is consistent with the understanding that such classical methods are often too conservative for the large-scale problems considered in this study.

**Real Data Experiments.** We compared the `proposed` and `w/o-pp` in terms of power, for the `cv` pipeline on eight real-world datasets from the UCI Machine Learning Repository (all licensed under the CC BY 4.0; see Appendix D.5 for more details). These experiments were conducted under the implicit assumption that features selected by the feature selection pipeline are truly relevant. This assumption is reasonable because both the `proposed` and `w/o-pp` evaluated in this study have been shown to control the type I error rate. From each original dataset, we randomly generated 1,000 sub-sampled datasets with sample sizes of $n \in \{100, 150, 200\}$. We then applied both the `proposed` and `w/o-pp` to assess their powers. The results, presented in Table 1, demonstrate that the `proposed` method has much higher power than the `w/o-pp` across all datasets for all sample sizes. Furthermore, the power of the `proposed` increases with increasing sample size $n$.

## 7. Conclusions

In this study, we introduced a novel framework for testing the statistical significance of feature selection pipelines in linear models, comprising multiple MVI, OD, and FS algorithms based on the concept of SI. Our long-term goal extends beyond this current study to ensure the reproducibil-

ity of data-driven decision-making by accounting for the entire pipeline from raw data to final results, with this study on a class of feature selection pipelines serving as a proof of concept. To achieve this future goal, there are still limitations on the applicable data analysis components, presenting several challenges in extending the proposed framework to more complex data analysis pipelines. Additionally, it is interesting to consider extending this framework to scenarios where data analysis pipelines are automatically constructed using state-of-the-art AutoML approaches.

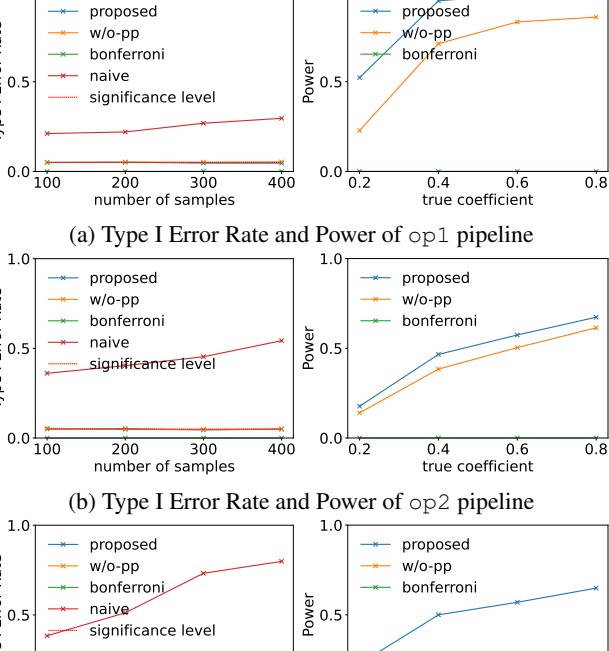

(a) Type I Error Rate and Power of `op1` pipeline

(b) Type I Error Rate and Power of `op2` pipeline

(c) Type I Error Rate and Power of `cv` pipeline

*Figure 3.* Type I Error Rate when changing the number of samples (left side) and Power when changing the true coefficient (right side). The proposed method (`proposed`), the ablation study (`w/o-pp`), and the Bonferroni method (`bonferroni`) successfully control the type I error rate across all settings and pipeline types. Among the methods that control the type I error rate, the proposed method has the highest power across all settings and pipeline types.

## Acknowledgements

This work was partially supported by JST CREST (JPMJCR21D3, JPMJCR22N2), JST Moonshot R&D (JPMJMS2033-05), RIKEN Center for Advanced Intelligence Project, and RIKEN Junior Research Associate Program.

## Impact Statement

This work, which focuses on statistical tests for feature selection pipelines, aims to enhance the reliability of AI and has the potential to broadly influence the machine learning community. On the other hand, it does not present significant ethical concerns or foreseeable societal consequences because this work is theoretical and, as of now, has no direct applications that might impact society or ethical considerations.

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

# A. Pipeline Components

## A.1. Missing-Value Imputation (MVI) Algorithm Components

A MVI algorithm component is represented as

$$f_{\mathrm{MVI}} : \{X, \boldsymbol{y}, \mathcal{O}, \mathcal{M}\} \mapsto \{X, \boldsymbol{y}^{(+)}, \mathcal{O}, \mathcal{M}\},$$

where $\boldsymbol{y} \in \mathbb{R}^{n'}$ is the response vector which excludes missing values and $\boldsymbol{y}^{(+)} \in \mathbb{R}^n$ is the vector with imputed missing values. MVI algorithms in this paper are *linear* algorithm in the sense that, using a matrix $D_X \in \mathbb{R}^{n \times n'}$ that depends on $X$ are written as $\boldsymbol{y}^{(+)} = D_X \boldsymbol{y}$.

**Mean Value Imputation.** This method replaces missing values with the mean value of observed data and allows for quick and easy imputation of missing values. An example of $D_X$ for $\boldsymbol{y} = (y_1, y_3, y_4)^\top$ (i.e., $y_2$ is missing value and $n = 4$) is:

$$D_X = \begin{pmatrix} 1 & 0 & 0 \\ 1/3 & 1/3 & 1/3 \\ 0 & 1 & 0 \\ 0 & 0 & 1 \end{pmatrix}.$$

**Nearest-Neighbor Imputation.** This method replaces missing values with the most similar instance in the dataset. In this method, similarity between instances is measured by some distance between their feature vectors. As distance measures $\ell(\cdot, \cdot)$, for example, Euclidean, Manhattan, or Chebyshev distance can be used. An example of $D_X$ for $\boldsymbol{y} = (y_1, y_3, y_4)^\top$ (i.e., $y_2$ is missing value and $n = 4$) is:

$$D_X = \begin{pmatrix} 1 & 0 & 0 \\ & \boldsymbol{e}_j^\top & \\ 0 & 1 & 0 \\ 0 & 0 & 1 \end{pmatrix}, \ j = \arg\min_{i \in \{1,3,4\}} \ell(\boldsymbol{x}_2, \boldsymbol{x}_i),$$

where $\boldsymbol{e}_j$ is the vector constructed by removing the indices of the missing values (i.e., $\{2\}$) from the $j$-th unit vector in $\mathbb{R}^4$.

**Regression Imputation.** This method replaces missing values with estimated values based on a regression model. We use the observed instances to estimate the regression coefficients, and then use the estimated coefficients to predict the missing values from its feature vector. We denote the indices of the missing values as NaN, and the indices of the observed values as $-$NaN. The regression coefficients can be estimated as $\hat{\boldsymbol{\beta}} = (X_{-\mathrm{NaN},:}^T X_{-\mathrm{NaN},:})^{-1} X_{-\mathrm{NaN},:}^T \boldsymbol{y}$ and then each imputed missing value $y_i^{(+)}, i \in \mathrm{NaN}$ can be expressed as $y_i^{(+)} = \boldsymbol{x}_i^\top \hat{\boldsymbol{\beta}}$. An example of $D_X$ for $\boldsymbol{y} = (y_1, y_3, y_4)^\top$ (i.e., $y_2$ is missing value and $n = 4$) is:

$$D_X = \begin{pmatrix} 1 & 0 & 0 \\ X_{\{2\},:}(X_{\{1,3,4\},:}^\top X_{\{1,3,4\},:})^{-1} X_{\{1,3,4\},:}^\top \\ 0 & 1 & 0 \\ 0 & 0 & 1 \end{pmatrix}$$

## A.2. Outlier Detection (OD) Algorithm Components

A OD algorithm component is represented as

$$f_{\mathrm{OD}} : \{X, \boldsymbol{y}^{(+)}, \mathcal{O}, \mathcal{M}\} \mapsto \{X, \boldsymbol{y}^{(+)}, \mathcal{O}', \mathcal{M}\},$$

where $\mathcal{O}'$ is the updated set of outliers.

**Cook's Distance-based OD.** This method identifies instances as outliers when *Cook's distance*, a measure of the influence of a particular instance on the entire regression model, exceeds a predefined threshold value. *Cook's distance* of the $i$-th instance is defined as

$$D_i = \frac{\sum_{j=1}^n \left(\hat{y}_j - \hat{y}_{j(i)}\right)^2}{d \, \mathrm{MSE}},$$

where $\hat{y}_j$ and $\hat{y}_{j(i)}$ are the predicted value of $j$-th instance from the regression model with and without $i$-th instance, respectively, and MSE is the mean squared error of the full model. This $D_i$ represents the standardized value of the change in predictions for all other instances due to the removal of $i$-th instance, and the larger $D_i$ is, the more it affects the model. By utilizing the leverage value, it can also be represented as

$$D_i = \frac{\hat{\varepsilon}_i^2}{d \, \text{MSE}} \frac{h_{ii}}{(1 - h_{ii})^2},$$

where $\hat{\varepsilon}_i$ is the $i$-th residual, $h_{ii}$ is the $i$-th leverage value (i.e., the diagonal component of the matrix $X(X^\top X)^{-1}X^\top$). We identify the $i$-th instance as an outlier if $D_i > \lambda$ where $\lambda$ is a predefined threshold value.

**DFFITS OD**    This method has the same concept as Cook's distance-based OD but uses *DFFITS* instead of Cook's distance as the measure of influence. *DFFITS* of the $i$-th instance is defined as

$$\text{DFFITS}_i = \frac{\hat{y}_i - \hat{y}_{i(i)}}{\sqrt{\text{MSE}_{(i)} h_{ii}}},$$

where $\hat{y}_i$ and $\hat{y}_{i(i)}$ are the predicted value of the $i$-th instance from the regression model with and without the $i$-th instance, respectively, $\text{MSE}_{(i)}$ is the mean squared error of the regression model without the $i$-th instance, and $h_{ii}$ is the $i$-th leverage value. Thus, *DFFITS* is a value that standardizes the difference between the predicted value when excluding and including a specific instance, and the larger $\text{DFFITS}_i$ is, the more it affects the model. By utilizing the external Studentized residual $r_{i,\text{ext}}$, it can also be represented as

$$\text{DFFITS}_i = \sqrt{\frac{h_{ii}}{1 - h_{ii}}} r_{i,\text{ext}}.$$

We identify the $i$-th instance as an outlier if $\text{DFFITS}_i^2 > \lambda d/(n-d)$ where $\lambda$ is a predefined threshold value and usually set to $4$.

$L_1$ **Regression based OD**    This method identifies instances as outliers by using $L1$ regularization for the mean-shift model. In this method, we assume that the unknown true value function $\boldsymbol{\mu}(X)$ follows the following mean-shift model:

$$\boldsymbol{\mu}(X) = X\boldsymbol{\beta} + \boldsymbol{u},$$

where $\boldsymbol{u} \in \mathbb{R}^n$ is an outlier term and $u_i \neq 0$ if the $i$-th instance is an outlier, otherwise $u_i = 0$. We estimate $(\hat{\boldsymbol{\beta}}_\lambda, \hat{\boldsymbol{u}}_\lambda)$ by solving the following optimization problem:

$$(\hat{\boldsymbol{\beta}}_\lambda, \hat{\boldsymbol{u}}_\lambda) = \underset{\boldsymbol{\beta} \in \mathbb{R}^d, \boldsymbol{u} \in \mathbb{R}^n}{\arg\min} \frac{1}{2n} \|\boldsymbol{y}^{(+)} - X\boldsymbol{\beta} - \boldsymbol{u}\|_2^2 + \lambda\|\boldsymbol{u}\|_1,$$

where $\lambda$ is a predefined regularization parameter. We identify the $i$-th instance as an outlier if $\hat{u}_{\lambda,i} \neq 0$.

### A.3. Feature Selection (FS) Algorithm Components

A FS algorithm component is represented as

$$f_{\text{FS}} : \{X, \boldsymbol{y}^{(+)}, \mathcal{O}, \mathcal{M}\} \mapsto \{X, \boldsymbol{y}^{(+)}, \mathcal{O}, \mathcal{M}'\},$$

where, $\mathcal{M}'$ is the updated set of features.

**Marginal Screening**    This method selects the $k$ features that are most correlated with the response variable, where $k$ is a predefined number. The correlation is computed as the absolute value of the inner product $|\boldsymbol{x}_j^\top \boldsymbol{y}^{(+)}|$ between the normalized feature vector $\boldsymbol{x}_j$ and the response vector $\boldsymbol{y}^{(+)}$.

**Stepwise Feature Selection**    This method selects features by iterating through the steps of adding or deleting the features that best improve the goodness of fit of the regression model. In this paper, we deal with forward stepwise feature selection, which only adds features. The residual sum of squares (RSS) of the least squares regression model constructed using the features selected up to the previous step is used as the goodness of fit of the model. First, a null model (a model consisting

of an intercept term) is used as an initial state, and in each step, RSS is calculated from the least squares regression model constructed with the features selected in the previous step and the residual of $\boldsymbol{y}^{(+)}$. After that, select the feature that minimize RSS and update the model. The algorithm terminates if the RSS is not improved by adding any feature, or if the number of selected features reaches a predefined upper limit.

**Lasso**  This method selects features by using a linear regression model with $L1$ regularization. We estimate the regression coefficient $\hat{\boldsymbol{\beta}}$ by solving the following optimization problem:

$$\hat{\boldsymbol{\beta}} = \underset{\boldsymbol{\beta} \in \mathbb{R}^d}{\arg\min} \frac{1}{2n} \|\boldsymbol{y}^{(+)} - X\boldsymbol{\beta}\|_2^2 + \lambda \|\boldsymbol{\beta}\|_1,$$

where $\lambda$ is a predefined regularization parameter. We select the features for which $\hat{\beta}_i \neq 0$.

## B. Proofs

### B.1. Proof of Theorem 3.1

According to the conditioning on $\mathcal{Q}_{\boldsymbol{Y}} = \mathcal{Q}_{\boldsymbol{y}}$, we have

$$\mathcal{Q}_{\boldsymbol{Y}} = \mathcal{Q}_{\boldsymbol{y}} \Leftrightarrow \left(I_{n'} - \frac{\boldsymbol{\eta}^\top \boldsymbol{\eta}}{\|\boldsymbol{\eta}\|^2}\right) \boldsymbol{Y} = \mathcal{Q}_{\boldsymbol{y}} \Leftrightarrow \boldsymbol{Y} = \boldsymbol{a} + \boldsymbol{b}z,$$

where $z = T(\boldsymbol{Y}) \in \mathbb{R}$. Then, we have

$$\begin{aligned}
&\{\boldsymbol{Y} \in \mathbb{R}^{n'} \mid \mathcal{M}_{\boldsymbol{Y}} = \mathcal{M}_{\boldsymbol{y}}, \mathcal{O}_{\boldsymbol{Y}} = \mathcal{O}_{\boldsymbol{y}}, \mathcal{Q}_{\boldsymbol{Y}} = \mathcal{Q}_{\boldsymbol{y}}\} \\
=&\{\boldsymbol{Y} \in \mathbb{R}^{n'} \mid \mathcal{M}_{\boldsymbol{Y}} = \mathcal{M}_{\boldsymbol{y}}, \mathcal{O}_{\boldsymbol{Y}} = \mathcal{O}_{\boldsymbol{y}}, \boldsymbol{Y} = \boldsymbol{a} + \boldsymbol{b}z, z \in \mathbb{R}\} \\
=&\{\boldsymbol{a} + \boldsymbol{b}z \in \mathbb{R}^{n'} \mid \mathcal{M}_{\boldsymbol{a}+\boldsymbol{b}z} = \mathcal{M}_{\boldsymbol{y}}, \mathcal{O}_{\boldsymbol{a}+\boldsymbol{b}z} = \mathcal{O}_{\boldsymbol{y}}, z \in \mathbb{R}\} \\
=&\{\boldsymbol{a} + \boldsymbol{b}z \in \mathbb{R}^{n'} \mid z \in \mathcal{Z}\}.
\end{aligned}$$

Therefore, we obtain

$$T(\boldsymbol{Y}) \mid \{\mathcal{M}_{\boldsymbol{Y}} = \mathcal{M}_{\boldsymbol{y}}, \mathcal{O}_{\boldsymbol{Y}} = \mathcal{O}_{\boldsymbol{y}}, \mathcal{Q}_{\boldsymbol{Y}} = \mathcal{Q}_{\boldsymbol{y}}\} \sim \mathrm{TN}(\boldsymbol{\eta}^\top \boldsymbol{\mu}, \sigma^2 \|\boldsymbol{\eta}\|^2, \mathcal{Z}).$$

### B.2. Proof of Theorem 3.2

By probability integral transformation, under the null hypothesis, we have

$$p_{\text{selective}} \mid \{\mathcal{M}_{\boldsymbol{Y}} = \mathcal{M}_{\boldsymbol{y}}, \mathcal{O}_{\boldsymbol{Y}} = \mathcal{O}_{\boldsymbol{y}}, \mathcal{Q}_{\boldsymbol{Y}} = \mathcal{Q}_{\boldsymbol{y}}\} \sim \mathrm{Unif}(0, 1),$$

which leads to

$$\mathbb{P}_{\mathrm{H}_0} \left(p_{\text{selective}} \leq \alpha \mid \mathcal{M}_{\boldsymbol{Y}} = \mathcal{M}_{\boldsymbol{y}}, \mathcal{O}_{\boldsymbol{Y}} = \mathcal{O}_{\boldsymbol{y}}, \mathcal{Q}_{\boldsymbol{Y}} = \mathcal{Q}_{\boldsymbol{y}}\right) = \alpha, \ \forall \alpha \in (0, 1).$$

For any $\alpha \in (0, 1)$, by marginalizing over all the values of the nuisance parameters, we obtain

$$\begin{aligned}
&\mathbb{P}_{\mathrm{H}_0} \left(p_{\text{selective}} \leq \alpha \mid \mathcal{M}_{\boldsymbol{Y}} = \mathcal{M}_{\boldsymbol{y}}, \mathcal{O}_{\boldsymbol{Y}} = \mathcal{O}_{\boldsymbol{y}}\right) \\
=&\int_{\mathbb{R}^{n'}} \mathbb{P}_{\mathrm{H}_0} \left(p_{\text{selective}} \leq \alpha \mid \mathcal{M}_{\boldsymbol{Y}} = \mathcal{M}_{\boldsymbol{y}}, \mathcal{O}_{\boldsymbol{Y}} = \mathcal{O}_{\boldsymbol{y}}, \mathcal{Q}_{\boldsymbol{Y}} = \mathcal{Q}_{\boldsymbol{y}}\right) \\
&\qquad \mathbb{P}_{\mathrm{H}_0} \left(\mathcal{Q}_{\boldsymbol{Y}} = \mathcal{Q}_{\boldsymbol{y}} \mid \mathcal{M}_{\boldsymbol{Y}} = \mathcal{M}_{\boldsymbol{y}}, \mathcal{O}_{\boldsymbol{Y}} = \mathcal{O}_{\boldsymbol{y}}\right) d\mathcal{Q}_{\boldsymbol{y}} \\
=&\alpha \int_{\mathbb{R}^{n'}} \mathbb{P}_{\mathrm{H}_0} \left(\mathcal{Q}_{\boldsymbol{Y}} = \mathcal{Q}_{\boldsymbol{y}} \mid \mathcal{M}_{\boldsymbol{Y}} = \mathcal{M}_{\boldsymbol{y}}, \mathcal{O}_{\boldsymbol{Y}} = \mathcal{O}_{\boldsymbol{y}}\right) d\mathcal{Q}_{\boldsymbol{y}} = \alpha.
\end{aligned}$$

Therefore, we also obtain

$$\begin{aligned}
&\mathbb{P}_{\mathrm{H}_0}(p_{\text{selective}} \leq \alpha) \\
=&\sum_{\mathcal{M}_{\boldsymbol{y}} \in 2^{[p]}} \sum_{\mathcal{O}_{\boldsymbol{y}} \in 2^{[n]}} \mathbb{P}_{\mathrm{H}_0}(\mathcal{M}_{\boldsymbol{y}}, \mathcal{O}_{\boldsymbol{y}}) \mathbb{P}_{\mathrm{H}_0} \left(p_{\text{selective}} \leq \alpha \mid \mathcal{M}_{\boldsymbol{Y}} = \mathcal{M}_{\boldsymbol{y}}, \mathcal{O}_{\boldsymbol{Y}} = \mathcal{O}_{\boldsymbol{y}}\right) \\
=&\alpha \sum_{\mathcal{M}_{\boldsymbol{y}} \in 2^{[p]}} \sum_{\mathcal{O}_{\boldsymbol{y}} \in 2^{[n]}} \mathbb{P}_{\mathrm{H}_0}(\mathcal{M}_{\boldsymbol{y}}, \mathcal{O}_{\boldsymbol{y}}) = \alpha.
\end{aligned}$$

## B.3. Proof of Theorem 4.1

It is sufficient to consider only $z$ as input to Algorithm 1. In addition, as a notation, we define $\mathcal{G}_i$ as the mapping that returns the last four components of $B_i$ for $i \in \{0, 1, \ldots, |V|\}$, i.e.,

$$\mathcal{G}_i \colon \mathbb{R} \ni z \mapsto (\mathcal{M}_{\boldsymbol{a}+\boldsymbol{b}z}^i, \mathcal{O}_{\boldsymbol{a}+\boldsymbol{b}z}^i, l_z^i, u_z^i) \in 2^{[p]} \times 2^{[n]} \times \mathbb{R}^2, \ i \in \{0, 1, \ldots, |V|\}$$

According to the above notation, all we have to show is that $\mathcal{G}_{|V|}(z) = \mathcal{G}_{|V|}(r)$ for any $z \in \mathbb{R}$ and any $r \in [l_z^{|V|}, u_z^{|V|}]$. We show this by mathematical induction.

In the case $i = 0$, it is obvious from the definition of $B_0$ in Algorithm 1 that $\mathcal{G}_0(z) = \mathcal{G}_0(r) = ([p], \emptyset, -\infty, \infty)$ for any $z \in \mathbb{R}$ and any $r \in [l_z^0, u_z^0] = [-\infty, \infty]$.

Next, we assume that for any fixed $i \in \{0, \ldots, |V| - 1\}$, $\mathcal{G}_j(z) = \mathcal{G}_j(r)$ for any $j \in \{0, \ldots, i\}$, any $z \in \mathbb{R}$ and any $r \in [l_z^j, u_z^j]$. Under this assumption, noting that $\mathrm{pa}(i+1) \subset \{0, \ldots, i\}$ from a property of topological sort, it is obvious that $\mathcal{G}_{i+1}(z) = \mathcal{G}_{i+1}(r)$ for any $z \in \mathbb{R}$ and any $r \in [l_z^{i+1}, u_z^{i+1}]$ from the update rule of $v_{i+1}$ described in §4.3.

## C. Details of the Update Rules

**Update Rulu for the Node of MVI.**   The node of MVI imputes the missing values in the response vector $\boldsymbol{a} + \boldsymbol{b}z$. All MVI algorithms considered in this study are expressed as linear transformations determined on the basis of $X$. Thus, let $D_X$ be the linear transformation matrix, the update rule should be as follows:

$$(X, \boldsymbol{a}, \boldsymbol{b}, z, \mathcal{M}, \mathcal{O}, l, u) \mapsto (X, D_X\boldsymbol{a}, D_X\boldsymbol{b}, z, \mathcal{M}, \mathcal{O}, l, u).$$

**Update Rule for the Node of FS.**   The node of FS selects the features $\mathcal{M}'(z)$ from the dataset $(X_{-\mathcal{O},\mathcal{M}}, \boldsymbol{a}_{-\mathcal{O}} + \boldsymbol{b}_{-\mathcal{O}}z)$, which means that feature selection is performed on the dataset extracted from $(X, \boldsymbol{a} + \boldsymbol{b}z)$ based on $\mathcal{M}$ and $\mathcal{O}$. For all FS algorithms considered in this study, the computation procedure to obtain the interval $[l_z, u_z] \ni z$, which satisfies

$$\forall r \in [l_z, u_z], \ \mathcal{M}'(r) = \mathcal{M}'(z),$$

have been proposed in previous studies (Lee & Taylor, 2014; Tibshirani et al., 2016; Lee et al., 2016). Utilizing this, the update rule should be as follows:

$$(X, \boldsymbol{a}, \boldsymbol{b}, z, \mathcal{M}, \mathcal{O}, l, u) \mapsto (X, \boldsymbol{a}, \boldsymbol{b}, z, \mathcal{M} \cap \mathcal{M}'(z), \mathcal{O}, \max(l, l_z), \min(u, u_z)).$$

**Update Rule for the Node of OD.**   The node of OD detects the outliers $\mathcal{O}'(z)$ from the dataset $(X_{-\mathcal{O},\mathcal{M}}, \boldsymbol{a}_{\mathcal{M}} + \boldsymbol{b}_{\mathcal{M}}z)$, which means that outlier detection is performed on the dataset extracted from $(X, \boldsymbol{a} + \boldsymbol{b}z)$ based on $\mathcal{M}$ and $\mathcal{O}$. For all OD algorithms considered in this study, the computation procedure to obtain the interval $[l_z, u_z] \ni z$, which satisfies

$$\forall r \in [l_z, u_z], \ \mathcal{O}'(r) = \mathcal{O}'(z),$$

have been proposed in previous studies (Chen & Bien, 2020). Utilizing this, the update rule should be as follows:

$$(X, \boldsymbol{a}, \boldsymbol{b}, z, \mathcal{M}, \mathcal{O}, l, u) \mapsto (X, \boldsymbol{a}, \boldsymbol{b}, z, \mathcal{M}, \mathcal{O} \cap \mathcal{O}'(z), \max(l, l_z), \min(u, u_z)).$$

**Update Rule for the Node of Union/Intersection of Features/Outliers.**   The node computes the union or intersection of selected features or detected outliers. With $E$ being the number of input edges, for each selected feature and detected outlier, the update rules should be as follows:

$$\{(X, \boldsymbol{a}, \boldsymbol{b}, z, \mathcal{M}_e, \mathcal{O}, l_e, u_e)\}_{e\in[E]} \mapsto (X, \boldsymbol{a}, \boldsymbol{b}, z, \sum_{e\in[E]} \mathcal{M}_e, \mathcal{O}, \max_{e\in[E]} l_e, \min_{e\in[E]} u_e),$$

$$\{(X, \boldsymbol{a}, \boldsymbol{b}, z, \mathcal{M}, \mathcal{O}_e, l_e, u_e)\}_{e\in[E]} \mapsto (X, \boldsymbol{a}, \boldsymbol{b}, z, \mathcal{M}, \sum_{e\in[E]} \mathcal{O}_e, \max_{e\in[E]} l_e, \min_{e\in[E]} u_e),$$

where $\sum$ represents the union or intersection depending on the type of the node.

# D. Details of the Experiments

## D.1. Additional Type I Error Rate Results

We also conducted experiments to investigate the type I error rate when the number of features $d$ is changed, and for the high-dimensional regression setting (i.e., where $d \gg n$). For the former case, we changed the number of features $d \in \{10, 20, 30, 40\}$ and set the number of samples $n$ to 200. For the latter case, we set the number of samples $n$ to 100 and changed the number of features $d \in \{400, 800, 1200, 1600\}$. It should be noted that, within this experimental setting, the op2 and op3 pipelines are used, to handle the high-dimensional regression setting. The op3 pipeline is defined by reversing the order of the $L_1$ regression-based outlier detection (OD) node and the marginal screening feature selection (FS) node in the op1 pipeline. In both cases, we generated the null datasets in the same way as in the main experiments (§6), and the results are shown in Figure 4 and Figure 5, respectively.

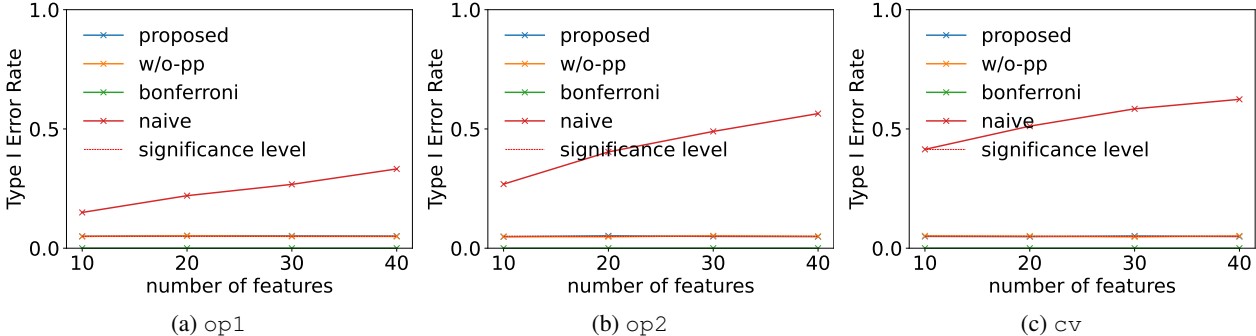

(a) op1        (b) op2        (c) cv

*Figure 4.* Type I Error Rate when changing the number of features $d$. Our proposed method (proposed), the ablation study (w/o-pp), and the Bonferroni method (bonferroni) successfully control the type I error rate across all settings and pipeline types.

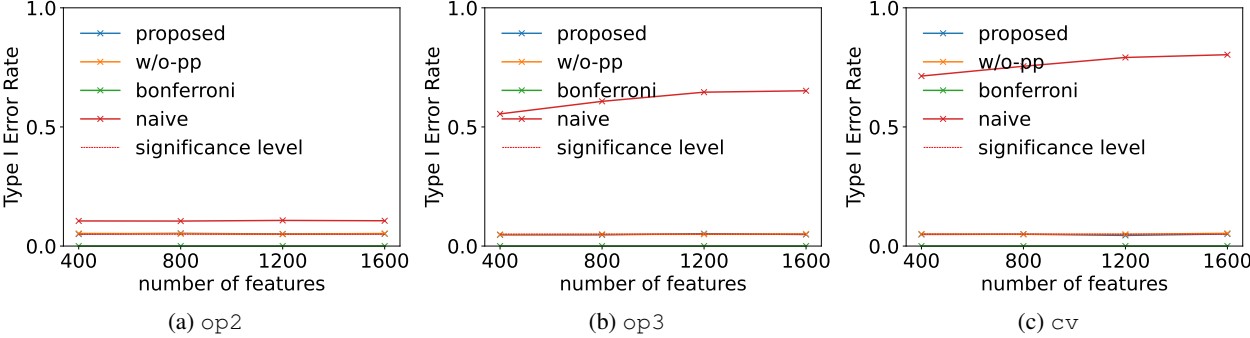

(a) op2        (b) op3        (c) cv

*Figure 5.* Type I Error Rate for the high-dimensional regression setting. Our proposed method (proposed), the ablation study (w/o-pp), and the Bonferroni method (bonferroni) successfully control the type I error rate across all settings and pipeline types.

## D.2. Effect of Missing Value Probability

We also conducted experiments to investigate the effect of the missing value probability on the type I error rate and power of the proposed method. In the experiments, we change the missing value probability $\rho \in \{0.03, 0.12, 0.21, 0.30\}$. For the type I error rate, we set the number of samples $n = 200$ and the number of features $d = 20$. For the power, we set the number of samples $n = 200$, the number of features $d = 20$, and the true coefficients $\Delta = 0.4$. In both cases, we generated the datasets in the same way as in the main experiments (§6), and the results are shown in Figure 6.

## D.3. Computational Time of the Proposed Method

We also conducted experiments to investigate the computational time of our proposed method by applying it to three types of pipeline structures (Default, Parallel, and Serial) with large-scale datasets. Default pipeline correspond to the op1 pipeline

in §6. Parallel and Serial pipelines are defined as in Figure 8, which clarifies the difference from the Default with components colored in pink. In the experiments, we change the number of samples $n \in \{400, 800, 1200, 1600\}$ with the number of features $d = 80$ and the number of features $d \in \{40, 80, 120, 160\}$ with the number of samples $n = 800$ to generate the null datasets in the same way as in the main experiments (§6). Note that in this experiment, we recorded the computational time for a single hypothesis testing (i.e., calculate one $p$-value) on a single CPU core. The results are shown in Figure 7.

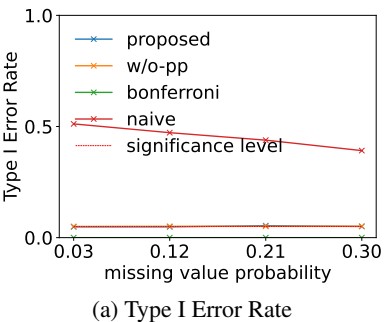

(a) Type I Error Rate

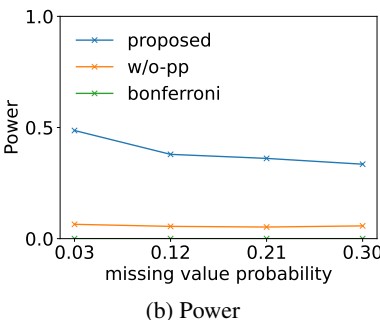

(b) Power

*Figure 6.* Type I Error Rate and Power when changing the missing value probability. The proposed method (`proposed`), the ablation study (`w/o-pp`), and the Bonferroni method (`bonferroni`) successfully control the type I error rate across all settings. Among the methods that control the type I error rate, the proposed method has the highest power across all settings.

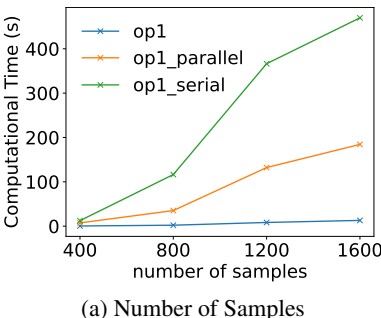

(a) Number of Samples

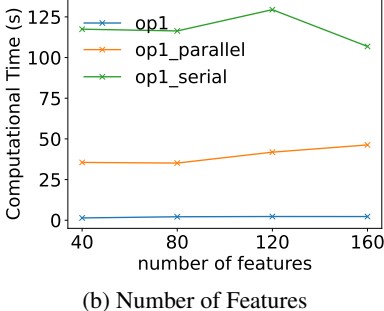

(b) Number of Features

*Figure 7.* Computational Time when changing the number of samples and features. The results show that computational time is exponentially increased as the number of samples increases while the number of features has no obvious effect. Moreover, it seems that increasing the number of nodes in the pipeline increases the computational time, but how much it increases also depends on the structure.

### D.4. Computer Resources

All numerical experiments were conducted on a computer with a 96-core 3.60GHz CPU and 512GB of memory.

### D.5. Details of the Real Datasets

We used the following eight real datasets from the UCI Machine Learning Repository. All datasets are licensed under the CC BY 4.0 license.

- Airfoil Self-Noise (Brooks et al., 1989) for Data1

- Concrete Compressive Strength (Yeh, 1998) for Data2

- Energy Efficiency (Tsanas & Xifara, 2012) for Data3 (heating load) and Data4 (cooling load)

- Gas Turbine CO and NOx Emission Data Set (gas, 2019) for Data5

- Real Estate Valuation (Yeh, 2018) for Data6

- Wine Quality (Cortez et al., 2009) for Data7 (red wine) and Data8 (white wine)

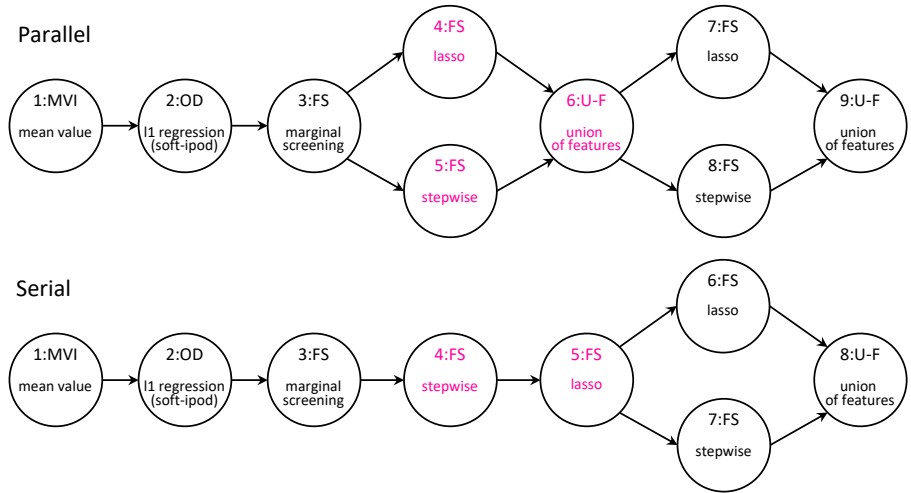

*Figure 8.* Definition of Parallel and Serial pipelines used in Figure 7.

# E. Robustness of Type I Error Rate Control

In this experiment, we confirmed the robustness of the proposed method for `cv` pipeline in terms of type I error rate control by applying our method to the two cases: the case where the variance is estimated from the same data and the case where the noise is non-Gaussian.

## E.1. Estimated Variance

In the case where the variance is estimated from the same data, we considered the same two options as in type I error rate experiments in §6 and Appendix D.1; number of samples and number of features. For each setting, we generated 10,000 null datasets $(X, \boldsymbol{y})$, where $X_{ij} \sim \mathcal{N}(0,1)$, $\forall(i,j) \in [n] \times [d]$ and $\boldsymbol{y} \sim \mathcal{N}(0, I_n)$ and estimated the variance $\hat{\sigma}^2$ as

$$\hat{\sigma}^2 = \frac{1}{n-d}\|\boldsymbol{y} - X(X^{\top}X)^{-1}X^{\top}\boldsymbol{y}\|_2^2.$$

We considered the three significance levels $\alpha = 0.05, 0.01, 0.10$. The results are shown in Figure 9 and our proposed method can properly control the type I error rate.

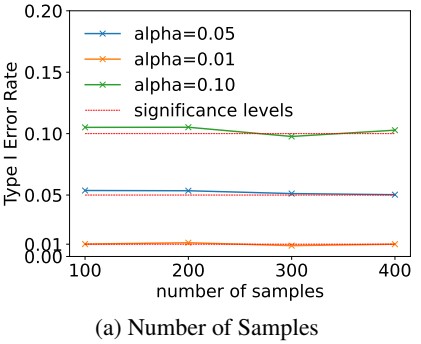

(a) Number of Samples

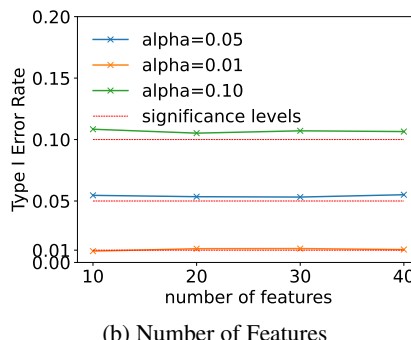

(b) Number of Features

*Figure 9.* Robustness of Type I Error Rate Control. Our proposed method can robustly control the type I error rate even when the variance is estimated from the same data.

## E.2. Non-Gaussian Noise

In the case where the noise is non-Gaussian, we set $n = 200$ and $d = 20$. As non-Gaussian noise, we considered the following five distribution families:

- `skewnorm`: Skew normal distribution family.

- `exponnorm`: Exponentially modified normal distribution family.

- `gennormsteep`: Generalized normal distribution family (limit the shape parameter $\beta$ to be steeper than the normal distribution, i.e., $\beta < 2$).

- `gennormflat`: Generalized normal distribution family (limit the shape parameter $\beta$ to be flatter than the normal distribution, i.e., $\beta > 2$).

- `t`: Student's t distribution family.

Note that all of these distribution families include the Gaussian distribution and are standardized in the experiment.

To conduct the experiment, we first obtained a distribution such that the 1-Wasserstein distance from $\mathcal{N}(0,1)$ is $l$ in each distribution family, for $l \in \{0.01, 0.02, 0.03, 0.04\}$. We then generated 10,000 null datasets $(X, \boldsymbol{y})$, where $X_{ij} \sim \mathcal{N}(0,1)$, $\forall(i,j) \in [n] \times [d]$ and $\boldsymbol{y}_i$, $\forall i \in [n]$ follows the obtained distribution. We considered the two significance levels $\alpha = 0.05, 0.01$. The results are shown in Figure 10 and our proposed method can properly control the type I error rate.

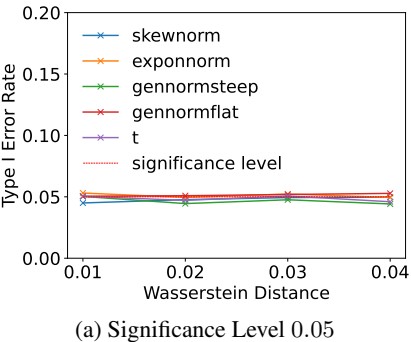
(a) Significance Level 0.05

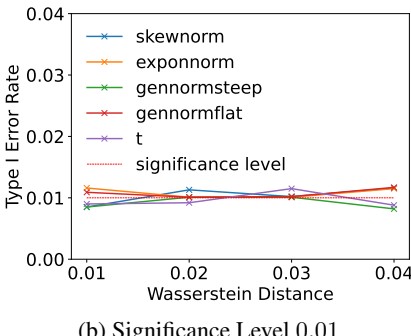
(b) Significance Level 0.01

*Figure 10.* Robustness of Type I Error Rate Control. Our proposed method can robustly control the type I error rate even when the noise follows non-Gaussian distributions.

## F. Automatic Pipeline Construction based on Cross-Validation

In this section, we discuss cross-validation for pipelines. We consider selecting the pipeline $\mathcal{P}$ from a given set of candidates $\{\mathcal{P}_1, \ldots, \mathcal{P}_S\}$ where $S$ is the number of candidates. Note that this formulation is general enough to handle many cross-validation targets in a unified form. For examples, (i) the case where only changing the regularization strength of lasso node, (ii) the case where changing the method of missing value imputation, and (iii) the case where changing the all structure of the pipeline (i.e., type and order of nodes).

Thereafter, we discuss how statistical inference changes when cross-validation is performed and how cross-validation can be formulated. Then, based on above discussion, Algorithm 1 is extended to be applicable to the case of cross-validation.

### F.1. Statistical Inference after Cross-Validation

**Changes in Statistical Inference** As a formulation of statistical inference after cross-validation, the discussion in §2 and §3 can be done in exactly the same way, except with two changes: (i) the procedure for computing $\mathcal{M}$ and $\mathcal{O}$ (in §2 and §3, $\mathcal{M}$ and $\mathcal{O}$ are simply the outputs of a given mapping $\mathcal{P}$ representing a target pipeline), and (ii) the dependence on the response vector $\boldsymbol{y}$ of which method to use for missing value imputation. This implies that the procedure for computing the truncation intervals $\mathcal{Z}$ in §4 can not be directly applied to the case of cross-validation.

**Formulation of Cross-Validation Procedure** We consider the case where $K$-fold cross-validation is performed. Let $(X, \boldsymbol{y})$ be the observed data set and $\{(T_k, V_k)\}_{k \in [K]}$ be the $K$ types of partition of training and validation sets, which

satisfies $T_k, V_k \in 2^{[n]}$, $T_k \cap V_k = \emptyset$, and $T_k \cup V_k = [n]$ for any $k \in [K]$. Then, the cross-validation error $E_s(X, \boldsymbol{y})$ for the pipeline $\mathcal{P}_s$ is defined as

$$E_s(X, \boldsymbol{y}) = \sum_{k \in [K]} \frac{1}{|V_k|} \|(D_X^s \boldsymbol{y})_{V_k} - X_{V_k, \mathcal{M}_{s,k}} \hat{\boldsymbol{\beta}}_{s,k}(\boldsymbol{y})\|_2^2,$$

where $D_X^s$ is the linear transformation matrix in the missing value imputation of the pipeline $\mathcal{P}_s$, $\hat{\boldsymbol{\beta}}_{s,k}(\boldsymbol{y}) = \left( X_{T_k \backslash \mathcal{O}_{s,k}, \mathcal{M}_{s,k}}^\top X_{T_k \backslash \mathcal{O}_{s,k}, \mathcal{M}_{s,k}} \right)^{-1} X_{T_k \backslash \mathcal{O}_{s,k}, \mathcal{M}_{s,k}}^\top (D_X^s \boldsymbol{y})_{T_k \backslash \mathcal{O}_{s,k}}$, and $(\mathcal{M}_{s,k}, \mathcal{O}_{s,k})$ is the output of the pipeline $\mathcal{P}_s$ with input $(X_{T_k}, (D_X^s \boldsymbol{y})_{T_k})$. In $K$-fold cross-validation, the pipeline $\mathcal{P}_{s^*}$ is selected to minimize the cross-validation error $E_s(X, \boldsymbol{y})$, i.e., $s^* = \arg\min_{s \in [S]} E_s(X, y)$.

### F.2. Auto-Conditioning for Cross-Validation

To conduct the statistical inference after cross-validation, it is suffice to have the procedure to compute the interval $[L_z, U_z]$ for any $z \in \mathbb{R}$ which satisfy

$$\forall r \in [L_z, U_z],$$
$$\arg\min_{s \in [S]} E_s(X, \boldsymbol{a} + \boldsymbol{b}r) = \arg\min_{s \in [S]} E_s(X, \boldsymbol{a} + \boldsymbol{b}z)(:= s(z)),$$
$$\mathcal{P}_{s(z)}(X, \boldsymbol{a} + \boldsymbol{b}r) = \mathcal{P}_{s(z)}(X, \boldsymbol{a} + \boldsymbol{b}z).$$

If we have this procedure, for any $r \in [L_z, U_z]$, the selected features and the detected outliers after selecting the pipeline by cross-validation from the data set $(X, \boldsymbol{a} + \boldsymbol{b}r)$ are invariant. Therefore, the $p_{\text{selective}}$ can be computed in exactly the same way as in §4 only by adding the condition $D_X^{s(z)} = D_X^{s^*}$ as well as the condition $\mathcal{M}_{\boldsymbol{a}+\boldsymbol{b}z} = \mathcal{M}_{\boldsymbol{y}}$ and $\mathcal{O}_{\boldsymbol{a}+\boldsymbol{b}z} = \mathcal{O}_{\boldsymbol{y}}$. Hereafter, we provide the above procedure by extending Algorithm 1.

For implementation of the above procedure, we compute two intervals $[L_z^{\text{cv}}, U_z^{\text{cv}}]$ and $[L_z^{\text{sel}}, U_z^{\text{sel}}]$ for any $z \in \mathbb{R}$ which satisfy

$$\forall r \in [L_z^{\text{cv}}, U_z^{\text{cv}}], \ \arg\min_{s \in [S]} E_s(X, \boldsymbol{a} + \boldsymbol{b}r) = \arg\min_{s \in [S]} E_s(X, \boldsymbol{a} + \boldsymbol{b}z)(:= s(z)),$$
$$\forall r \in [L_z^{\text{sel}}, U_z^{\text{sel}}], \ \mathcal{P}_{s(z)}(X, \boldsymbol{a} + \boldsymbol{b}r) = \mathcal{P}_{s(z)}(X, \boldsymbol{a} + \boldsymbol{b}z),$$

respectively, and let $L_z = \max(L_z^{\text{cv}}, L_z^{\text{sel}})$ and $U_z = \min(U_z^{\text{cv}}, U_z^{\text{sel}})$.

To compute the interval $[L_z^{\text{cv}}, U_z^{\text{cv}}]$, we use Algorithm 1 repeatedly. For any $(s, k) \in [S] \times [K]$ and any $z \in \mathbb{R}$, we compute the interval $[L_z^{(s,k)}, U_z^{(s,k)}]$ which satisfy

$$\forall r \in [L_z^{(s,k)}, U_z^{(s,k)}], \ \mathcal{P}_s(X_{T_k}, (D_X^s \boldsymbol{a} + D_X^s \boldsymbol{b}r)_{T_k}) = \mathcal{P}_s(X_{T_k}, (D_X^s \boldsymbol{a} + D_X^s \boldsymbol{b}z)_{T_k}),$$

by using Algorithm 1 with input $(\mathcal{P}_s, X_{T_k}, (D_X^s \boldsymbol{a} + D_X^s \boldsymbol{b}z)_{T_k}, z)$. Thus, if we consider the $k$-th term of the sum in $E_s(X, \boldsymbol{a} + \boldsymbol{b}r)$ as a function of $r$, then it becomes quadratic in $r$ on the interval $[L_z^{(s,k)}, U_z^{(s,k)}]$. Therefore, on the interval $\cap_{s \in [S]} \cap_{k \in [K]} [L_z^{(s,k)}, U_z^{(s,k)}]$, the cross-validation errors $\{E_s(X, \boldsymbol{a} + \boldsymbol{b}r)\}_{s \in [S]}$ are all quadratic in $r$. On this interval $\cap_{s \in [S]} \cap_{k \in [K]} [L_z^{(s,k)}, U_z^{(s,k)}]$, the simultaneous inequalities for $r$

$$E_{s(z)}(X, \boldsymbol{a} + \boldsymbol{b}r) \le E_s(X, \boldsymbol{a} + \boldsymbol{b}r), \forall s \in [S],$$

with $s(z) = \arg\min_{s \in [S]} E_s(X, \boldsymbol{a} + \boldsymbol{b}z)$ become simultaneous quadratic inequalities, which can be solved analytically to finally obtain the interval $[L_z^{\text{cv}}, U_z^{\text{cv}}]$.

To compute the interval $[L_z^{\text{sel}}, U_z^{\text{sel}}]$, we simply use Algorithm 1 with input $(\mathcal{P}_{s(z)}, X, \boldsymbol{a}, \boldsymbol{b}, z)$.

## G. Examples of Implementations

We show an example of how the pipeline is implemented in our experiments. Listing 2 shows the implementation of the automatic pipeline construction scheme referred to as cv in the experiments (§6). Note that we can specify the candidates of the parameters for each operation and perform cross-validation to determine the optimal pipeline by using fit method.

*Listing 2.* Code example that defines the automatic pipeline construction scheme referred to as `cv` in the experiments. We can create an instance of manager class which handles identically structured pipelines, each with a different hyperparameter set, simply by specifying each operation and its candidates of parameters in turn (corresponding to `option1_multi` and `option2_multi`). Manager instances can use the OR operator `|` to create new manager instance which handles all of the pipelines that each instance handles. To perform hypothesis testing after cross-validation, we can call the `fit` and `inference` method of the manager instance sequentially.

```python
import numpy as np
from si4pipeline import *

def option1_multi() -> PipelineManager:
    X, y = initialize_dataset()
    y = mean_value_imputation(X, y)

    O = soft_ipod(X, y, [0.02, 0.018])
    X, y = remove_outliers(X, y, O)

    M = marginal_screening(X, y, [3, 5])
    X = extract_features(X, M)

    M1 = stepwise_feature_selection(X, y, [2, 3])
    M2 = lasso(X, y, [0.08, 0.12])
    M = union(M1, M2)
    return construct_pipelines(output=M)

def option2_multi() -> PipelineManager:
    X, y = initialize_dataset()
    y = definite_regression_imputation(X, y)

    M = marginal_screening(X, y, [3, 5])
    X = extract_features(X, M)

    O = cook_distance(X, y, [2.0, 3.0])
    X, y = remove_outliers(X, y, O)

    M1 = stepwise_feature_selection(X, y, [2, 3])
    M2 = lasso(X, y, [0.08, 0.12])
    M = intersection(M1, M2)
    return construct_pipelines(output=M)

manager = option1_multi() | option2_multi()
X, y = np.random.normal(size=(100, 10)), np.random.normal(size=100)

manager.tune(X, y, num_folds=2)
M, p_list = manager.inference(X, y, sigma=1.0)
```

