# OpenReview forum: "Statistical Test for Feature Selection Pipelines by Selective Inference"
_ICML.cc/2025/Conference — ICML 2025 oral_

### Official Review · Reviewer_sBGD · 2025-03-07

**Overall Recommendation:** 4

**Summary:**

This paper presents a statistical test to assess the significance of data analysis pipelines, which transform raw data by integrating various analysis algorithms. The paper specifically focuses on feature selection pipelines for linear models, which are composed by value imputation algorithms, outlier detection algorithms, and feature selection algorithms. The proposed statistical test builds upon the technique of selective inference, and it is theoretically shown that it can control the probability of false positive feature selection at any desired level. Experiments on synthetic and real data are provided to demonstrate the validity and effectiveness of the proposed statistical test.

## update after rebuttal ##
The rebuttal from the authors has supported my initial recommendation to accept the paper.

**Claims And Evidence:**

The paper provides convincing theoretical and experimental results to support the validity and effectiveness of the proposed statistical test.

**Essential References Not Discussed:**

The paper presents a summary of the most related works in selective inference and AutoML literature.

**Experimental Designs Or Analyses:**

Overall, the experimental design seems correct. However, there are certain details that I am unclear about:

1) The probability of a missing value was set to 0.03, which seems quite low. Is it possible that selecting such value to be low can somehow affect the performance of the statistical test?
2) I am not entirely sure I understood the real data experiments, and more specifically, why it is necessary to generate random datasets from each original dataset to illustrate the performance of the proposed test?
3) The real-world datasets considered have very few features (<15), so I am wondering how the proposed test will behave in settings with larger number of features?

**Methods And Evaluation Criteria:**

The paper uses both synthetic and real-wold datasets to assess the performance of the statistical test. In addition, it considers three baselines, a version of the proposed test without parametric programming, a classical z-test, and the Bonferroni correction. In general, the proposed methods and evaluation criteria make sense for the problem at hand. Since the focus of the paper seems to be on feature selection pipelines for the moment, would not it make sense to compare with any of the baselines below?

- Knockoff-based feature selection
- Meinshausen, Nicolai, and Peter Bühlmann. "Stability selection." Journal of the Royal Statistical Society Series B: Statistical Methodology 72, no. 4 (2010): 417-473.

**Other Comments Or Suggestions:**

- Line 186: remove "is" from "Making the p-value is a random variable"
- Line 246: replace "can" with "are" in "Note that DAGs can.."
- Line 310: replace "currently" with "current"
- Line 312: replace "details" with "details"
- The sentence "Missing values .. of 0.03" is repeated twice in the Experimental Setup section.
- In Section 6, "Methods for Comparison", for op1 and op2, I believe that the reference should be to Figure 1, not Figure 2.

**Other Strengths And Weaknesses:**

Strengths:
(+) Well written paper that builds upon selective inference to assess statistical significance of data analysis pipelines.
(+) Includes experiments with both synthetic and real-world data
(+) The figures provided facilitate the understanding of the proposed approach.

Weaknesses:
(-) Some parts of the experimental study are unclear.

**Questions For Authors:**

I have already stated these questions, but I restate them below:

1) The probability of a missing value was set to 0.03, which seems quite low. How (or better does) such value can affect the performance of the statistical test?
2) I am not entirely sure I understood the real data experiments, and more specifically, why it is necessary to generate random datasets from each original dataset to illustrate the performance of the proposed test?
3) The real-world datasets considered have very few features (<15), so I am wondering how the proposed test will behave in settings with larger number of features.
4) Since the focus of the paper seems to be on feature selection pipelines for the moment, would not it make sense to compare with any of the baselines below?
- Knockoff-based feature selection
- Meinshausen, Nicolai, and Peter Bühlmann. "Stability selection." Journal of the Royal Statistical Society Series B: Statistical Methodology 72, no. 4 (2010): 417-473.

**Relation To Broader Scientific Literature:**

The proposed statistical test builds upon the selective inference statistical technique to provide a new way of assessing the significance of data analysis pipelines. The specific focus is on feature selection pipelines for linear models.

**Theoretical Claims:**

The proofs of the theoretical claims in the paper seem correct.

---

> ### Author Rebuttal · Authors · 2025-03-31
>
> We thank the reviewer for your feedback.
>
> > The probability of a missing value was set to 0.03, which seems quite low. How (or better does) such value can affect the performance of the statistical test?
>
> This probability was set only for experimental convenience. The validity of the proposed method, specifically its control of the Type I Error Rate, is unaffected by how missing values are present in the response vector. However, while a higher proportion of missing values naturally leads to a decrease in statistical power, the proposed method maintains its advantage over the comparison methods.
>
> > I am not entirely sure I understood the real data experiments, and more specifically, why it is necessary to generate random datasets from each original dataset to illustrate the performance of the proposed test?
>
> We need multiple datasets to evaluate the performances of statistical tests (type I error rate and power). Therefore, for experiments on real datasets, we created multiple datasets by randomly resampling the original dataset. Such an evaluation is a standard approach for assessing statistical tests on real datasets.
>
> > The real-world datasets considered have very few features (<15), so I am wondering how the proposed test will behave in settings with larger number of features.
>
> The validity of the proposed method and its superior statistical power compared to baseline methods remain consistent even as the number of features increases. Please refer to the results of the synthetic data experiments on high-dimensional data provided in the appendix (see Figure 5 in Appendix D).
>
> > Since the focus of the paper seems to be on feature selection pipelines for the moment, would not it make sense to compare with any of the baselines below?
> > - Knockoff-based feature selection
> > - Meinshausen, Nicolai, and Peter Bühlmann. "Stability selection." Journal of the Royal Statistical Society Series B: Statistical Methodology 72, no. 4 (2010): 417-473.
>
> Methods such as Knockoff-based feature selection and Stability Selection are designed to control the False Discovery Rate (FDR), defined as the expected proportion of falsely selected features within the selected set. In contrast, our proposed method performs hypothesis testing on individual features within a selected feature set. Because they address different inferential goals (set-level FDR vs. individual feature significance), the methods are not directly comparable.

---

> > ### Comment · Reviewer_sBGD · 2025-04-03
> >
> > Thank you for the responses provided to my questions. I have some follow up clarifications:
> >
> > 1. I appreciate the reply regarding the probability of missing value. However, I argue that there are no results in the paper that suggest that the proposed method maintains its advantage over the comparison methods when this value is increased. Adding even a small experiment in the camera-ready version can strengthen the statement you provided.
> >
> > 2. As a future recommendation, a study that involves a large number of real-world datasets with high number of features could be beneficial to support the importance of the particular statistical test.
> >
> > 3. My question regarding the low number of features was specifically for real-world datasets, not synthetic ones, since the former ones include variability in conditions, not necessarily present in synthetic datasets.

---

> > > ### Author Response · Authors · 2025-04-05
> > >
> > > We thank the reviewer for your further feedback.
> > >
> > > > I appreciate the reply regarding the probability of missing value. However, I argue that there are no results in the paper that suggest that the proposed method maintains its advantage over the comparison methods when this value is increased. Adding even a small experiment in the camera-ready version can strengthen the statement you provided.
> > >
> > > Thank you for your suggestion.
> > > Following the reviewer's recommendation, we conducted experiments comparing statistical power when varying probabilities of missing values.
> > > The results, presented in the table below, confirm the advantage of our proposed method even when the probability of missing values increases (the experimental setup is described later).
> > > We will add these results to the revised manuscript.
> > >
> > > | Probability of Missing Values | 0.03 | 0.12 | 0.21 | 0.30 |
> > > |---|---|---|---|---|
> > > | proposed | **0.487** | **0.379** | **0.361** | **0.335** |
> > > | w/o-pp | 0.064 | 0.055 | 0.052 | 0.057 |
> > > | bonferroni | 0.000 | 0.000 | 0.000 | 0.000 |
> > >
> > > > As a future recommendation, a study that involves a large number of real-world datasets with high number of features could be beneficial to support the importance of the particular statistical test.
> > >
> > > > My question regarding the low number of features was specifically for real-world datasets, not synthetic ones, since the former ones include variability in conditions, not necessarily present in synthetic datasets.
> > >
> > > Thank you for your suggestion; we had initially misinterpreted your feedback.
> > > We also plan to apply our testing framework to more practical high-dimensional data analysis tasks, such as gene expression data analysis, in the future.
> > >
> > > ---
> > >
> > > * The experimental setup for the above table is as follows:
> > >     - We used the `cv` pipeline for the experiments.
> > >     - We set the number of samples $n$ to $200$, the number of features $d$ to $20$, and the true coefficient $\Delta$ to $0.4$.
> > >     - We changed the probability of missing values from $\\{0.03, 0.12, 0.21, 0.30\\}$.
> > >     - We generated 10,000 datasets in the same way as in the main experiments (Section 6).

---

### Official Review · Reviewer_LDea · 2025-03-10

**Overall Recommendation:** 4

**Summary:**

The authors propose an extension of selective inference techniques from single procedures (lasso, marginal screening) to pipelines. They develop a statistical test that they claim to have better power for data analysis pipelines with multiple, data-adaptive decision points.

**Claims And Evidence:**

The claims of the work are clearly stated and supported mostly by theorems. The data-based evidence relies on a few somewhat out-dated datasets, but is not central to the contribution of the paper. The authors appropriately cast it as having somewhat limited applicability (linear models with normal errors, specific feature selection steps), but with an eye towards generalizing to autoML pipelines, which is a good goal.

**Essential References Not Discussed:**

None noted

**Experimental Designs Or Analyses:**

As noted above, the only caveat with the experiments are that the datasets are somewhat out-dated. It is kind of an ancillary point to the main value of the paper.

**Methods And Evaluation Criteria:**

Yes, the proposed methods make sense. The evaluation criteria is also sensible. I like that the authors focused on power since a method like this may be prone to over-conditioning and power loss.

**Other Comments Or Suggestions:**

None

**Other Strengths And Weaknesses:**

The paper is clearly written and well reasoned. The value proposition is clear. To me it appears to be a good extension of prior work.

The main weaknesses I see are:

1. It is not clear how well this procedure could truly generalize, if at all, to more complicated ML models or pipelines. The strong assumptions that are made about error normality and a fixed design matrix seem to be quite central to the validity of the statistic.
2. Line search is computationally expensive, as evidenced in the appendix fig 6, and for these simple datasets is already quite slow.

**Questions For Authors:**

Could the authors say more about how this prototype is expected to be extendable to more complicated pipelines and datasets?

**Relation To Broader Scientific Literature:**

The paper is related to selective inference literature, and extends it from single procedures to pipelines. The paper seems clear about its contribution in this regard.

**Theoretical Claims:**

I checked proofs 3.1, 3.2, and 4.1, although I could have missed some details. Each of them seems clear.

- 3.1 uses a standard selective inference trick to transform $Y$ and obtain the truncated normal distribution $T(Y)$
- 3.2 is similarly aligned with selective inference theory to establish the uniformity of the p value
- 4.1 uses some parametric programming arguments to define when selected features and detected outliers remain unchanged (i.e. when the update rules do not trigger a change)

The proofs appear to correctly apply techniques seen in related works (Lee et al 2016, Tibshirani et al 2016).

---

> ### Author Rebuttal · Authors · 2025-03-31
>
> We thank the reviewer for your feedback.
>
> > It is not clear how well this procedure could truly generalize, if at all, to more complicated ML models or pipelines. The strong assumptions that are made about error normality and a fixed design matrix seem to be quite central to the validity of the statistic.
>
> > Could the authors say more about how this prototype is expected to be extendable to more complicated pipelines and datasets?
>
> Conditional selective inference was initially developed targeting the problem of feature selection in linear models, but it has recently been applied to various problems such as clustering and anomaly detection. In this paper, as a proof of concept, we consider a feature selection pipeline for linear models, but we believe that the same concept can be applied to a wide range of other problems as well. All the analysis components considered in this paper are those for which the selection events can be characterized by sets of linear or quadratic inequalities. In addition to the nine specific analysis components in the paper, there are many other analysis components whose selection events can be characterized within this framework. Technicaly, it is easy to incorporate them as additional components of our current pipeline framework.
>
> > Line search is computationally expensive, as evidenced in the appendix fig 6, and for these simple datasets is already quite slow.
>
> Although the line search is computationally expensive, we expect that parallelization offers potential for significant time reduction. The vertical axis in Figure 6 shows the computation time needed for a single hypothesis test (i.e., calculate one p-value) on a single CPU core. Our preliminary experiments indicate that using 16 cores reduces this time to roughly one-fifth. We will add an explanation of this in the revised manuscript.

---

### Official Review · Reviewer_ewmH · 2025-03-22

**Overall Recommendation:** 4

**Summary:**

This paper presents a statistical testing framework based on selective inference (SI) for assessing the significance of features selected through full feature selection pipelines. These pipelines may include steps such as missing value imputation, outlier detection, and feature selection. The key idea is to compute valid p-values by conditioning on the fact that the features were selected through a specific sequence of data-dependent operations. The proposed framework is modular and can accommodate a range of commonly used pipeline components. Empirical results on both synthetic and real datasets show that the method effectively controls Type I error and reduces false discoveries compared to standard, naive testing approaches—particularly when selection is driven by the data.

**Claims And Evidence:**

Yes. The claims are supported by both theoretical arguments and empirical evidence. The experimental section demonstrates that naive approaches can yield misleading significance results, while the proposed method performs consistently under various setups.

**Essential References Not Discussed:**

None identified.

**Experimental Designs Or Analyses:**

I have carefully looked at the experimental section in the main part of the paper. The design is clear, comparisons are appropriate, and the analysis effectively demonstrates the strengths of the proposed method.

**Methods And Evaluation Criteria:**

Yes. The methods and evaluation criteria are appropriate for the problem. The use of synthetic data for controlled studies and real datasets for practical illustration is well-justified.

**Other Comments Or Suggestions:**

Line 110: "AD – anomaly detection?" → consider changing to OD - "outlier detection" for consistency with terminology used elsewhere.

**Other Strengths And Weaknesses:**

Strengths:

1. The application of selective inference to this setting is a compelling and timely contribution.

2. The framework appears practical, modular, and generalizable to many pipelines used in practice.

3. The paper is well written, and the problem motivation is very clear.

Weaknesses:

1. Figures (especially Figure 3) could be improved to make differences between methods more visually accessible.

**Questions For Authors:**

1. Figure 3: It is difficult to distinguish between the performance of various methods, as many lines are overlapping. Consider using different linestyles, colors, or markers to more clearly show the differences.

**Relation To Broader Scientific Literature:**

This paper extends recent work in post-selection inference to the setting of feature selection pipelines, where rigorous statistical testing is often lacking. Its main contribution is the development of a framework that supports modular pipelines—composed of multiple data-dependent preprocessing and selection steps—while still enabling valid p-value computation for selected features. This represents a meaningful advance beyond prior work focused on single-step selection procedures.

**Theoretical Claims:**

I had a quick look at the proof of Theorem 3.1, which characterizes the distribution of the test statistic under the selective inference framework. The derivation appears correct and follows known techniques in selective inference literature. The modular treatment of pipeline components is a nice extension of this theory.

---

> ### Author Rebuttal · Authors · 2025-03-31
>
> We thank the reviewer for your feedback.
>
> > Line 110: "AD – anomaly detection?" → consider changing to OD - "outlier detection" for consistency with terminology used elsewhere.
>
> Thank you for pointing this out. We change the term to OD in the revised manuscript.
>
> > Figure 3: It is difficult to distinguish between the performance of various methods, as many lines are overlapping. Consider using different linestyles, colors, or markers to more clearly show the differences.
>
> We improve the figure to make it easier to distinguish between the different methods in the revised manuscript.

---

### Decision · Program_Chairs · 2025-05-01

**Decision:**

Accept (oral)

**Comment:**

All reviewers had a very positive impression of this paper: The underlying research question was considered interesting and of high practical relevance, proposed concept of selective inference seems to be well-motivated, clearly described and technically sound, the experiments are (mostly) convincing. There were some minor points of criticism concerning some details of the experimental study or the style of some figures, and one potentially more significant critical comment concerning the need for strong statistical assumptions in the theoretical analysis. In summary, however, to me the positive aspects clearly outweigh the negative ones, so I recommend acceptance of this paper.